# KRONECKER-FACTORED CURVATURE APPROXIMATIONS FOR RECURRENT NEURAL NETWORKS

**James Martens**
DeepMind
`jamesmartens@google.com`

**Jimmy Ba**
Department of Computer Science
University of Toronto
Toronto, Canada
`jimmy@psi.toronto.edu`

**Matthew Johnson**
Google Brain
`mattjj@google.com`

## ABSTRACT

Kronecker-factor Approximate Curvature (Martens & Grosse, 2015) (K-FAC) is a 2nd-order optimization method which has been shown to give state-of-the-art performance on large-scale neural network optimization tasks (Ba et al., 2017). It is based on an approximation to the Fisher information matrix (FIM) that makes assumptions about the particular structure of the network and the way it is parameterized. The original K-FAC method was applicable only to fully-connected networks, although it has been recently extended by Grosse & Martens (2016) to handle convolutional networks as well. In this work we extend the method to handle RNNs by introducing a novel approximation to the FIM for RNNs. This approximation works by modelling the statistical structure between the gradient contributions at different time-steps using a chain-structured linear Gaussian graphical model, summing the various cross-moments, and computing the inverse in closed form. We demonstrate in experiments that our method significantly outperforms general purpose state-of-the-art optimizers like SGD with momentum and Adam on several challenging RNN training tasks.

## 1 INTRODUCTION

As neural networks have become ubiquitous in both research and applications the need to efficiently train has never been greater. The main workhorses for neural net optimization are stochastic gradient descent (SGD) with momentum and various 2nd-order optimizers that use diagonal curvature-matrix approximations, such as RMSprop (Tieleman & Hinton, 2012) and Adam (Ba & Kingma, 2015). While the latter are typically easier to tune and work better out of the box, they unfortunately only offer marginal performance improvements over *well-tuned* SGD on most problems.

Because modern neural networks have many millions of parameters it is computationally too expensive to compute and invert an entire curvature matrix and so approximations are required. While early work on non-diagonal curvature matrix approximations such as TONGA (Le Roux et al., 2008) and the Hessian-free (HF) approach (Martens, 2010; Martens & Sutskever, 2011; 2012; Desjardins et al., 2013; Sainath et al., 2013) demonstrated the potential of such methods, they never achieved wide adoption due to issues of scalability (to large models in the case of the former, and large datasets in the case of the latter).

Motivated in part by these older results and by the more recent success of centering and normalization methods (e.g. Schraudolph, 1998; Vatanen et al., 2013; Ioffe & Szegedy, 2015) a new family of methods has emerged that are based on non-diagonal curvature matrix approximations the rely on the special structure of neural networks. Such methods, which include Kronecker-factored approximated curvature (K-FAC) (Martens & Grosse, 2015), Natural Neural Nets (Desjardins et al., 2015), Practical Riemannian Neural Networks (Marceau-Caron & Ollivier, 2016), and others (Povey

et al., 2015), have achieved state-of-the-art optimization performance on various challenging neural network training tasks and benchmarks.

While the original K-FAC method is applicable only to standard feed-forward networks with fully connected layers, it has recently been extended to handle convolutional networks (Grosse & Martens, 2016) through the introduction of the "Kronecker Factors for Convolution" (KFC) approximation. Ba et al. (2017) later developed a distributed asynchronous version which proposed additional approximations to handle very large hidden layers.

In this work we develop a new family of curvature matrix approximations for recurrent neural networks (RNNs) within the same design space. As in the original K-FAC approximation and the KFC approximation, we focus on the Fisher information matrix (a popular choice of curvature matrix), and show how it can be approximated in different ways through the adoption of various approximating assumptions on the statistics of the network's gradients. Our main novel technical contribution is an approximation which uses a chain-structured linear Gaussian graphical model to describe the statistical relationship between gradient contributions coming from different time-steps. Somewhat remarkably, it is possible to sum the required cross-moments to obtain a Fisher approximations which has enough special algebraic structure that it can still be efficiently inverted. In experiments we demonstrate the usefulness of our approximations on several challenging RNN training tasks.

## 2 NOTATION AND BACKGROUND

### 2.1 NETWORK, LOSS, AND OBJECTIVE FUNCTION

We denote by $f(x, \theta)$ the neural network function associated evaluated on input $x$, where $\theta$ are the parameters. We will assume a loss function of the form $L(y, z) = -\log r(y|z)$, where $r$ is the density function associated with a predictive distribution $R$. The loss associated with a single training case is then given by $L(y, f(x, \theta)) \equiv -\log r(y|f(x, \theta))$. Throughout the rest of this document we will use the following special notation for derivatives of the single-case loss w.r.t. some arbitrary variable $Z$ (possibly matrix-valued):

$$\mathcal{D}Z = \frac{\mathrm{d}L(y, f(x, \theta))}{\mathrm{d}Z}$$

The objective function which we wish to minimize is the expected loss $h(\theta) = \mathbb{E}_Q[L(y, f(x, \theta))]$ over the training distribution $Q$ on $x$ and $y$.

### 2.2 THE FISHER, THE NATURAL GRADIENT, AND 2ND-ORDER OPTIMIZATION

The Fisher information matrix (aka "the Fisher") associated with the *model's predictive distribution* $P_{y|x}(\theta)$ is given by

$$F = \mathbb{E}\left[\mathcal{D}\theta\mathcal{D}\theta^\top\right] = \mathrm{cov}(\mathcal{D}\theta, \mathcal{D}\theta).$$

Note that here, and for the remainder of this paper, $y$ is taken to be distributed according to the model's predictive conditional distribution $P_{y|x}(\theta)$, so that $\mathbb{E}[\mathcal{D}Z] = 0$ for any variable $Z$ that is conditionally independent of $y$ given the value of $f(x, \theta)$ (this includes $\mathcal{D}\theta$). All expectations and covariances are defined accordingly. This is done because the expectation that defines the Fisher information matrix uses $P_{y|x}(\theta)$. If we were to instead use the training distribution $Q_{y|x}$ on $y$, we would essentially be computing to the "empirical Fisher" (or approximations thereof), which as argued by Martens (2014) is a less appropriate choice for a curvature matrix than the true Fisher.

The natural gradient is defined as $F^{-1}\nabla h$, and is the update direction used in natural gradient descent. As argued by Amari (1998), natural gradient descent has the two key advantages: it is invariant to the parameterization of the model, and has "Fisher efficient" convergence[1]. However, as shown by Martens (2014) these two facts have several important caveats. First, the parameterization invariance only holds approximately in practice when non-infinitesimal step-sizes are used. Second, Fisher efficiency is actually a weak property possessed by simpler methods like SGD with

---

[1]Roughly speaking, this means that it converges at the asymptotically optimal rate (with optimal constant) for *arbitrary* statistical estimation procedures (as a function of the amount of samples from $Q$ observed).

Polyak/parameter averaging (Polyak & Juditsky, 1992), and even then will only be achieved when the method converges to a global minimizer *and* the model is capable of perfectly capturing the true distribution of $y$ given $x$.

An alternative explanation for the empirical success of the natural gradient method is that it is a 2nd-order method, whose update minimizes the following local quadratic approximation to the objective $h(\theta + \delta)$:

$$\frac{1}{2}\delta^\top F\delta + \nabla h(\theta)^\top \delta + h(\theta). \tag{1}$$

This is similar to the 2nd-order Taylor series approximation of $h(\theta + \delta)$, but with the Fisher substituted in for the Hessian. This substitution can be justified by the observation that the Fisher is a kind of PSD approximation to the Hessian (Pascanu & Bengio, 2014; Martens, 2014). And as argued by Martens (2014), while stochastic 2nd-order methods like natural gradient descent cannot beat the asymptotically optimal Fisher efficient convergence achieved by SGD with Polyak averaging, they can enjoy better *pre-asymptotic* convergence rates. Moreover, insofar as gradient noise can be mitigated through the use of large mini-batches – so that stochastic optimization starts to resemble deterministic optimization – the theoretical advantages of 2nd-order methods become further pronounced, which agrees with the empirical observation that the use of large-minibatches speeds up 2nd-methods much more than 1st-order methods (Martens & Grosse, 2015; Ba et al., 2017).

In addition to providing an arguably better theoretical argument for the success of natural gradient methods, their interpretation as 2nd-order methods also justifies the common practice of computing the natural gradient as $(F + \lambda I)^{-1}\nabla h$ instead of $F^{-1}\nabla h$. In particular, this practice can be viewed as a type of "update damping/regularization", where one encourages $\delta$ to lie within some region around $\delta = 0$ where eqn. 1 remains a trustworthy approximation (e.g. Nocedal & Wright, 2006; Martens & Sutskever, 2012).

### 2.3 KRONECKER-FACTORED APPROXIMATE CURVATURE (K-FAC)

Because modern neural network have millions (or even billions) of parameters it is computationally too expensive to compute and invert the Fisher. To address this problem, the K-FAC method of Martens & Grosse (2015) uses a block-diagonal approximation of the Fisher (where the blocks correspond to entire layers/weight matrices), and where the blocks are further approximated as Kronecker products between much smaller matrices. The details of this approximation are given in the brief derivation below.

Let $W$ be a weight matrix in the network which computes the mapping

$$s = Wa,$$

where $a$ and $s$ are vector-valued inputs and outputs respectively and denote

$$g = \mathcal{D}s.$$

As in the original K-FAC paper we will assume that $a$ includes a homogeneous coordinate with value 1 so that the bias vector may be folded into the matrix $W$.

*Here and throughout the rest of this document, $F$ will refer to the block of the Fisher corresponding to this particular weight-matrix $W$.*

The Kronecker product of matrices $B$ and $C$, denoted by $B \otimes C$ for matrices $B \in \mathbb{R}^{m \times n}$ and $C$ of arbitrary dimensions, is a block matrix defined by

$$B \otimes C \equiv \begin{bmatrix} [B]_{1,1}C & \cdots & [B]_{1,n}C \\ \vdots & \ddots & \vdots \\ [B]_{m,1}C & \cdots & [B]_{m,n}C \end{bmatrix}$$

Note that the Kronecker product has many convenient properties that we will make use of in this paper. (See Van Loan (2000) for a good discussion of the Kronecker product and its properties.)

A simple application of the chain rule gives $\mathcal{D}W = ga^\top$. If we approximate $g$ and $a$ as statistically independent, we can write $F$ as

$$
\begin{aligned}
F &= \mathbb{E}[\text{vec}(\mathcal{D}W)\,\text{vec}(\mathcal{D}W)^\top] = \mathbb{E}[\text{vec}(ga^\top)\,\text{vec}(ga^\top)^\top] = \mathbb{E}[(a \otimes g)(a \otimes g)^\top] \\
&= \mathbb{E}[(aa^\top) \otimes (gg^\top)] = \mathbb{E}[aa^\top] \otimes \mathbb{E}[gg^\top] = A \otimes G,
\end{aligned}
$$

where we have defined

$$A = \mathbb{E}[aa^\top] \quad \text{and} \quad G = \mathbb{E}[gg^\top].$$

The matrices $A$ and $G$ can be estimated using simple Monte Carlo methods, and averaged over lots of data by taking an exponentially decaying average across mini-batches.

This is the basic Kronecker factored approximation (Heskes, 2000; Martens & Grosse, 2015; Povey et al., 2015), which is related to the approximation made in the Natural Neural Nets approach (Desjardins et al., 2015). It is shown by Martens & Grosse (2015) that the approximation is equivalent to neglecting the higher-order cumulants of the $a$s and $g$s, or equivalently, assuming that they are Gaussian distributed. To see why this approximation is useful, we observe that inversion and multiplication of a vector by $F$ amounts to inverting the factor matrices $A$ and $G$ and performing matrix-matrix multiplications with them, due to the following two basic identities:

$$(B \otimes C)^{-1} = B^{-1} \otimes C^{-1} \quad \text{and} \quad (B \otimes C)\operatorname{vec}(X) = \operatorname{vec}(CXB^\top). \tag{2}$$

The required inversion and matrix multiplication operations are usually computational feasible because the factor matrices have dimensions equal to the size of the layers, which is typically just a few thousand. And when they are not, additional approximations can be applied, such as approximate/iterative inversion (Povey et al., 2015), or additional Kronecker-factorization applied to either $A$ or $G$ (Ba et al., 2017). Moreover, the computation of the inverses can be amortized across iterations of the optimizer at the cost of introducing some staleness into the estimates.

## 3 APPROXIMATING $F$ FOR RNNS

The basic Kronecker-factored approximation to the Fisher block $F$ described in the previous section assumed that the weight matrix $W$ was used to compute a *single* mapping of the form $s = Wa$. When $W$ is used to compute multiple such mappings, as is often the case for RNNs, or a mapping of a different flavor, as is the case for convolutional networks (CNNs), the approximation is not applicable, strictly speaking.

Grosse & Martens (2016) recently showed that by making additional approximating assumptions, the basic Kronecker-factored approximation can be extended to convolutional layers. This new approximation, called "KFC", is derived by assuming that gradient contributions coming from different spatial locations are uncorrelated, and that their intra and inter-location statistics are spatially homogeneous, in the sense that they look the same from all reference locations. These assumptions are referred to "spatially uncorrelated derivatives" and "spatial homogeneity," respectively.

In this section we give the main technical contribution of this paper, which is a family of Kronecker-based approximations of $F$ that can be applied to RNNs. To build this we will apply various combinations of the approximating assumptions used to derive the original K-FAC and KFC approaches, along with several new ones, including an approximation which works by modelling the statistical structure between the gradient contributions from time-steps using a chain-structured linear Gaussian graphical model.

### 3.1 PRELIMINARIES

Let $W$ be some weight matrix which is used at $\mathcal{T}$ different time-steps (or positions) to compute the mapping

$$s_t = Wa_t,$$

where $t$ indexes the time-step. $\mathcal{T}$ is allowed to vary between different training cases.

Defining $g_t = \mathcal{D}s_t$, the gradient of the single-case loss with respect to $W$ can be written as

$$\mathcal{D}W = \sum_{t=1}^{\mathcal{T}} g_t a_t^\top = \sum_{t=1}^{\mathcal{T}} \mathcal{D}_t W,$$

where $\mathcal{D}_t W = g_t a_t^\top$ denotes the contribution to the gradient from time-step $t$. When it is more convenient to work with the vector-representations of the matrix-valued variables $\mathcal{D}_t W$ we will use the notation

$$w_t = \operatorname{vec}(\mathcal{D}_t W),$$

so that $\text{vec}(\mathcal{D}W) = \sum_{t=1}^{\mathcal{T}} w_t$.

Let $F_{\mathcal{T}}$ denote the conditional Fisher of $\mathcal{D}W$ for a particular value of $\mathcal{T}$. We have

$$F_{\mathcal{T}} = \mathbb{E}[\text{vec}(\mathcal{D}W)\,\text{vec}(\mathcal{D}W)^{\top}|\mathcal{T}] = \mathbb{E}\left[\left.\left(\sum_{t=1}^{\mathcal{T}} w_t\right)\left(\sum_{t=1}^{\mathcal{T}} w_t\right)^{\top}\right|\mathcal{T}\right] = \sum_{t=1}^{\mathcal{T}}\sum_{s=1}^{\mathcal{T}} \mathbb{E}\left[w_t w_s^{\top}|\mathcal{T}\right]. \tag{3}$$

Observe that $F$ can be computed from $F_{\mathcal{T}}$ via $F = \mathbb{E}_{\mathcal{T}}[F_{\mathcal{T}}]$.

To proceed with our goal of obtaining a tractable approximation to $F$ we will make several approximating assumptions, as discussed in the next section.

## 3.2 BASIC INITIAL APPROXIMATIONS

### 3.2.1 INDEPENDENCE OF $\mathcal{T}$

One simplifying approximation we will make immediately is that $\mathcal{T}$ is independent of the $w_t$'s, so that $\mathbb{E}[w_t w_s^{\top}|\mathcal{T}] = \mathbb{E}[w_t w_s^{\top}]$. In this case eqn. 3 can be written as

$$F_{\mathcal{T}} = \sum_{t=1}^{\mathcal{T}}\sum_{s=1}^{\mathcal{T}} \mathbb{E}[w_t w_s^{\top}] = \sum_{t=1}^{\mathcal{T}}\sum_{s=1}^{\mathcal{T}} V_{t,s}, \tag{4}$$

where we have defined $V_{t,s} = \mathbb{E}[w_t w_s^{\top}]$.

Independence of $\mathcal{T}$ and the $w_t$'s is a reasonable approximation assumption to make because 1) for many datasets $\mathcal{T}$ is constant (which formally implies independence), and 2) even when $\mathcal{T}$ varies substantially, shorter sequences will typically have similar statistical properties to longer ones (e.g. short paragraphs of text versus longer paragraphs).

### 3.2.2 TEMPORAL HOMOGENEITY

Another convenient and natural approximating assumption we will make is that the $w_t$'s are temporally homogeneous, which is to say that the statistical relationship between any $w_t$ and $w_s$ depends only on their distance in time ($d = t - s$). This is analogous to the "spatial homogeneity" assumption of KFC. Under this assumption the following single-subscript notation is well-defined: $V_{t-s} = \mathbb{E}[w_t w_s^{\top}]$. We note that $V_{-d} = V_d^{\top}$.

Applying this notation to eqn. 4 we have

$$F_{\mathcal{T}} = \sum_{t=1}^{\mathcal{T}}\sum_{s=1}^{\mathcal{T}} V_{t,s} = \sum_{d=-\mathcal{T}}^{\mathcal{T}} (\mathcal{T} - |d|)V_d = \sum_{d=0}^{\mathcal{T}} (\mathcal{T} - d)V_d + \sum_{d=0}^{\mathcal{T}} (\mathcal{T} - d)V_d^{\top} - \mathcal{T}I, \tag{5}$$

where we have used the fact that there are $\mathcal{T} - |d|$ ways to write $d$ as $t - s$ for $t, s \in \{1, 2, \ldots, \mathcal{T}\}$.

Temporal homogeneity is a pretty mild approximation, and is analogous to the frequently used "steady-state assumption" from dynamical systems. Essentially, it is the assumption that the Markov chain defined by the system "mixes" and reaches its equilibrium distribution. If the system has any randomness, and its external inputs reach steady-state, the steady-state assumption is quite accurate for states sufficiently far from the beginning of the sequence (which will be most of them).

### 3.2.3 INDEPENDENCE BETWEEN THE $a_t$'S AND THE $g_t$'S

If we have that $a_t$ and $g_s$ are pair-wise independent for each $t$ and $s$, which is the obvious generalization of the basic approximation used to derive the K-FAC approach, then following a similar derivation to the one from Section 2.3 we have

$$V_{t,s} = \mathbb{E}[(a_t a_s^{\top}) \otimes (g_t g_s^{\top})] = \mathbb{E}[(a_t a_s^{\top}) \otimes (\mathcal{D}s_t \mathcal{D}s_s^{\top})] = A_{t,s} \otimes G_{t,s},$$

where we have defined

$$A_{t,s} = \mathbb{E}[a_t a_s^{\top}] \quad \text{and} \quad G_{t,s} = \mathbb{E}[g_t g_s^{\top}].$$

Extending our temporal homogeneity assumption from the $w_t$'s to the $a_t$'s and $g_t$'s (which is natural to do since $w_t = \text{vec}(g_t a_t^\top)$), the following notation becomes well-defined:

$$A_{t-s} = A_{t,s} \quad \text{and} \quad G_{t-s} = G_{t,s},$$

which allows us to write

$$V_d = A_d \otimes G_d.$$

## 3.3 AN INITIAL ATTEMPT TO OBTAIN A TRACTABLE FISHER APPROXIMATION

Given the approximating assumptions made in the previous subsections we have

$$F_\mathcal{T} = \sum_{d=-\mathcal{T}}^{\mathcal{T}} (\mathcal{T} - |d|) V_d = \sum_{d=-\mathcal{T}}^{\mathcal{T}} (\mathcal{T} - |d|)(A_d \otimes G_d).$$

Assuming for the moment that all of the training sequences have the same length, so that $F = F_{\mathcal{T}_0}$ for some $\mathcal{T}_0$, we have that $F$ will be the sum of $2\mathcal{T}_0 + 1$ Kronecker products.

Without assuming any additional structure, such as a relationship between the various $A_d$'s or $G_d$'s, there doesn't appear to be any efficient way to invert such a sum. One can use the elementary identity $(B \otimes C)^{-1} = B^{-1} \otimes C^{-1}$ to invert a single Kronecker product, and there exists decomposition-based methods to efficiently invert sums of two Kronecker products (see Martens & Grosse (2015)), however there is no known *efficient* algorithm for inverting sums of three or more Kronecker products. Thus is appears that we must make additional approximating assumptions in order to proceed.

## 3.4 ASSUMING INDEPENDENCE OF THE $w_t$'S ACROSS TIME

If we assume that the contributions to the gradient (the $w_t$'s) are independent across time, or at least uncorrelated, this means that $V_d = 0$ for $d \neq 0$. This is analogous to the "spatially uncorrelated derivatives" assumption of KFC.

In this case eqn. 5 simplifies to

$$F_\mathcal{T} = \sum_{d=-\mathcal{T}}^{\mathcal{T}} (\mathcal{T} - |d|) V_d = (\mathcal{T} - 0) V_0 = \mathcal{T} V_0,$$

so that

$$F = \mathbb{E}_\mathcal{T}[F_\mathcal{T}] = \mathbb{E}_\mathcal{T}[\mathcal{T} V_0] = \mathbb{E}_\mathcal{T}[\mathcal{T}] V_0.$$

Using the identities in eqn. 2, and the symmetry of $A_0$ and $G_0$, we can thus efficiently multiply $F^{-1}$ by a vector $z = \text{vec}(Z)$ using the formula

$$F^{-1} z = \frac{1}{\mathbb{E}_\mathcal{T}[\mathcal{T}]} \text{vec}(G_0^{-1} Z A_0^{-1}). \tag{6}$$

This is, up to normalization by $\mathbb{E}_\mathcal{T}[\mathcal{T}]$, identical to the inverse multiplication formula used in the original K-FAC approximation for fully-connected layers.

We note that $\mathbb{E}_\mathcal{T}[\mathcal{T}] = \sum_i \omega_i \mathcal{T}_i$, where $\mathcal{T}_i$ are the different values of $\mathcal{T}$, and $\omega_i \geqslant 0$ are normalized weights (with $\sum_i \omega_i = 1$) that measure their proportions in the training set.

## 3.5 MODELING THE RELATIONSHIPS BETWEEN THE $w_t$'S USING AN LGGM

As we saw in Section 3.3, the approximation assumptions made in Section 3.2 (independence of $\mathcal{T}$, temporal homogeneity, and independence between the $a_t$'s and the $g_t$'s), aren't sufficient to yield a tractable formula for $F^{-1}$. And while additionally assuming independence across time of the $w_t$'s is sufficient (as shown in Section 3.4), it seems like an overly severe approximation to make.

In this section we consider a less severe approximation which we will show still produces a tractable $F^{-1}$. In particular, we will assume that the statistical relationship of the $w_t$'s is described by a simple linear Gaussian graphical model (LGGM) with a compact parameterization (whose size is independent of $\mathcal{T}$). Such an approach to computing a tractable Fisher approximations was first

explored by Grosse & Salakhutdinov (2015) for RBMs, although our use of it here is substantially different, and requires additional mathematical machinery.

The model we will use is a fairly natural one. It is a linear Gaussian graphical model with a one-dimensional chain structure corresponding to time. The graphical structure of our model is given by the following picture:

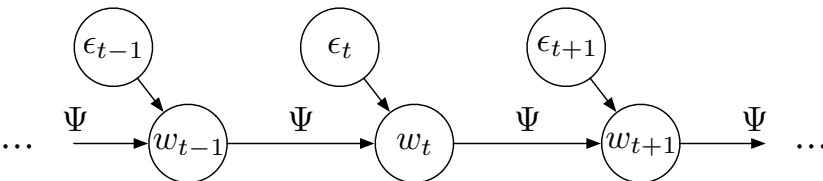

Variables in the model evolve forward in time according to the following equation:

$$w_t = \Psi w_{t-1} + \epsilon_t$$

where $\Psi$ is a square matrix and $\epsilon_t$ are i.i.d. from $\mathcal{N}(0, \Sigma)$ for some positive definite matrix $\Sigma$ (which is the conditional covariance of $w_t$ given $w_{t-1}$).

Due to the well-known equivalence between directed and undirected Gaussian graphical models for tree-structured graphs like this one, the decision of whether to make the edges directed or undirected, and whether to have them point forwards or backwards in time, are irrelevant from a modeling perspective (and thus to the Fisher approximation we eventually compute). We will use a directed representation purely for mathematical convenience.

We will assume that our model extends infinitely in both directions, with indices in the range $(-\infty, \infty)$, so that the $w_t$'s are all in their stationary distribution (with respect to time). For this to yield a well-defined model we require that $\Psi$ has spectral radius $< 1$.

The intuition behind this model structure is clear. The correlations between gradient contributions (the $w_t$'s) at two different time-steps should be reasonably well explained by the gradient contributions made at time-steps between them. In other words, they should be *approximately* Markovian.

We know that the gradient computations are generated by a process, Back-prop Through Time (BPTT), where information flows only between consecutive time-steps (forwards through time during the "forward pass", and backwards during the "backwards pass"). This process involves temporal quantities which are external to the $w_t$'s, such as the inputs $x$ and activations for other layers, which essentially act as "hidden variables". The evolution of these external quantities may be described by their own separate temporal dynamics (e.g. the unknown process which generates the true $x$'s), and thus the $w_t$'s won't be Markovian in general. But insofar as the $w_t$'s (or equivalently the $a_t$'s and $g_t$'s) encode the *relevant* information contained in these external variables, they should be approximately Markovian. (If they contained *all* of the information they would be exactly Markovian.)

A similar approximation across consecutive layers was made in the "block-tridiagonal" version of the original K-FAC approach. It was shown by Martens & Grosse (2015) that this approximation was a pretty reasonable one. The linear-Gaussian assumption meanwhile is a more severe one to make, but it seems necessary for there to be any hope that the required expectations remain tractable.

### 3.5.1 INITIAL COMPUTATIONS

Define the following "transformed" versions of $F_{\mathcal{T}}$ and $\Psi$:

$$\hat{F}_{\mathcal{T}} = V_0^{1/2} F_{\mathcal{T}} V_0^{1/2} \quad \text{and} \quad \hat{\Psi} = \hat{V}_1 = V_0^{-1/2} \Psi V_0^{1/2}.$$

As shown in Section A.1 of the appendix we have

$$\hat{F}_{\mathcal{T}} = \sum_{d=0}^{\mathcal{T}} (\mathcal{T} - d)\hat{\Psi}^d + \left( \sum_{d=0}^{\mathcal{T}} (\mathcal{T} - d)\hat{\Psi}^d \right)^{\top} - \mathcal{T}I$$
$$= \zeta_{\mathcal{T}}(\hat{\Psi}) + \zeta_{\mathcal{T}}(\hat{\Psi}^{\top}) - \mathcal{T}I \qquad (7)$$

where

$$\zeta_{\mathcal{T}}(x) = \frac{\mathcal{T}(1-x) - x(1-x^{\mathcal{T}})}{(1-x)^2}.$$

(Note that rational functions can be evaluated with matrix arguments in this way, as discussed in Section A.1.)

Our goal is to compute $\hat{F}^{-1}$, from which we can recover $F^{-1}$ via the simple relation $F^{-1} = V_0^{-1/2}\hat{F}^{-1}V_0^{-1/2}$.

Unfortunately it doesn't appear to be possible to simplify this formula sufficiently enough to allow for the efficient computation of $\hat{F}^{-1} = \mathbb{E}_{\mathcal{T}}[\hat{F}_{\mathcal{T}}]^{-1}$ when $\hat{\Psi}$ is a Kronecker product (which it will be when $V_0$ and $V_1$ are). The difficulty is due to both the appearance of $\hat{\Psi}$ and its transpose (which are not codiagonalizable/commutative in general), and various higher powers of $\hat{\Psi}$.

To proceed from this point and obtain a formula which can be efficiently evaluated when $\hat{\Psi}$ is a Kronecker product, we will make one of two simplifying assumptions/approximations, which we call "Option 1" and "Option 2" respectively. These are explained in the next two subsections.

### 3.5.2 OPTION 1: $V_1$ IS SYMMETRIC

If $V_1$ (the cross-moment over time) is symmetric, this implies that $\hat{\Psi} = \hat{V}_1 = V_0^{-1/2}V_1V_0^{-1/2}$ is also symmetric. Thus by eqn. 7 we have

$$\hat{F}_{\mathcal{T}} = \zeta_{\mathcal{T}}(\hat{\Psi}) + \zeta_{\mathcal{T}}(\hat{\Psi}) - \mathcal{T}I = \eta_{\mathcal{T}}(\hat{\Psi}),$$

where

$$\eta_{\mathcal{T}}(x) = 2\zeta_{\mathcal{T}}(x) - \mathcal{T} = \frac{2(\mathcal{T}(1-x) - x(1-x^{\mathcal{T}}))}{(1-x)^2} - \mathcal{T} = \frac{\mathcal{T}(1-x^2) - 2x(1-x^{\mathcal{T}})}{(1-x)^2}.$$

Let $U \operatorname{diag}(\hat{\psi})U^\top = \hat{\Psi}$ be the eigen-decomposition of $\hat{\Psi}$. By the above expression for $\hat{F}_{\mathcal{T}}$ we have that $\hat{F}_{\mathcal{T}} = U \operatorname{diag}(\eta_{\mathcal{T}}(\hat{\psi}))U^\top$, where $f(b)$ denotes the component-wise evaluation of a function $f$ for each component of the vector $b$, i.e. $[f(b)]_i = f([b]_i)$. We thus have

$$\hat{F} = \mathbb{E}_{\mathcal{T}}[\hat{F}_{\mathcal{T}}] = \mathbb{E}_{\mathcal{T}}[U \operatorname{diag}(\eta_{\mathcal{T}}(\hat{\psi}))U^\top] = U \operatorname{diag}(\mathbb{E}_{\mathcal{T}}[\eta_{\mathcal{T}}(\hat{\psi})])U^\top.$$

Inverting both sides of this yields

$$\hat{F}^{-1} = U \operatorname{diag}(\gamma(\hat{\psi}))U^\top \tag{8}$$

where we have defined $\gamma(x) = 1/\mathbb{E}_{\mathcal{T}}[\eta_{\mathcal{T}}(x)]$.

This expression can be efficiently evaluated when $\hat{\Psi}$ is a Kronecker product since the eigendecomposition of a Kronecker product can be easily obtained from the eigendecomposition of the factors. Evaluation of $\gamma(\hat{\psi})$ is done component-wise (i.e. $[\gamma(\hat{\psi})]_i = \gamma([\hat{\psi}]_i)$) and is thus easy to perform. See Section 3.5.5 for further details.

$V_1$ is symmetric if and only if $\hat{\Psi}$ is symmetric. And as shown in the proof of Proposition 1 (see Appendix A.1) $\hat{\Psi}$ has the interpretation of being the transition matrix of an LGGM which describes the evolution of "whitened" versions of the $w_t$'s (given by $\hat{w}_t = V_0^{-1/2}w_t$). Linear dynamical systems with symmetric transition matrices arise frequently in machine learning and related areas (Huang et al., 2016; Hazan et al., 2017), particularly because of the algorithmic techniques they enable. Intuitively, a symmetric transition matrix allows allows one to model exponential decay of different basis components of the signal over time, but not rotations between these components (which are required to model sinusoidal/oscillating signals).

Note that the observed/measured $V_1$ may or may not be exactly symmetric up to numerical precision, even if it well approximated as symmetric. For these calculations to make sense it must be exactly symmetric, and so even if it turns out to be approximately symmetric one should ensure that it is exactly so by using the symmetrized version $(V_1 + V_1^\top)/2$.

### 3.5.3 OPTION 2: COMPUTING THE LIMITING VALUE INSTEAD

If $V_1$ is not well approximated as symmetric, another option is to approximate

$$\hat{F} = \mathbb{E}_{\mathcal{T}}[\hat{F}_{\mathcal{T}}^{(\infty)}]$$

(instead of $\hat{F} = \mathbb{E}_{\mathcal{T}}[\hat{F}_{\mathcal{T}}]$), where we define

$$\hat{F}_{\mathcal{T}}^{(\infty)} \equiv \lim_{\mathcal{T}' \to \infty} \frac{\mathcal{T}}{\mathcal{T}'} \hat{F}_{\mathcal{T}'}.$$

This is essentially equivalent to the assumption that the training sequences are all infinitely long, which may be a reasonable one to make in practice. We re-scale by the factor $\frac{\mathcal{T}}{\mathcal{T}'}$ to achieve the proper scaling characteristics of $\hat{F}_{\mathcal{T}}$, and to ensure that the limit actually exists.

As shown in Section A.2 of the appendix this yields the following remarkably simple expression for $\hat{F}^{-1}$:

$$\hat{F}^{-1} = \frac{1}{\mathbb{E}_{\mathcal{T}}[\mathcal{T}]} (I - \hat{\Psi})(I - \hat{\Psi}^{\top}\hat{\Psi})^{-1}(I - \hat{\Psi}^{\top}). \tag{9}$$

Despite the fact that it includes both $\hat{\Psi}$ and $\hat{\Psi}^{\top}$, this formula can be efficiently evaluated when $\hat{\Psi}$ is a Kronecker product due to the existance of decomposition-based techniques for inverting matrices of the form $A \otimes B + C \otimes D$. See Section 3.5.5 for further details.

This approximation can break down if some of the linear components of $\hat{w}_t$ have temporal autocorrelations close to 1 (i.e. $[\hat{\psi}]_i \approx 1$ for some $i$) and $\mathcal{T}$ is relatively small. In such a case we will have that $[\hat{\psi}]_i^{\mathcal{T}}$ is large for some $i$ (despite being raised to the $\mathcal{T}$-th power) so that $\hat{F}_{\mathcal{T}}^{(\infty)}$ may essentially "overcount" the amount of temporal correlation that contributes to the sum.

This can be made more concrete by noting that the approximation is essentially equivalent to taking $\zeta_{\mathcal{T}}(x) \approx \lim_{\mathcal{T}' \to \infty} \frac{\mathcal{T}}{\mathcal{T}'} \zeta_{\mathcal{T}'}(x) \equiv \kappa(x)$ for each $x = [\hat{\psi}]_i$. We can express the error of this as

$$|\kappa(x) - \zeta_{\mathcal{T}}(x)| = \left| \frac{\mathcal{T}}{1-x} - \frac{\mathcal{T}(1-x) - x(1-x^{\mathcal{T}})}{(1-x)^2} \right| = \left| \frac{x(1-x^{\mathcal{T}})}{(1-x)^2} \right|.$$

It is easy to see how this expression, when evaluated at $x = [\hat{\psi}]_i$, might be large when $[\hat{\psi}]_i$ is close to 1, and $\mathcal{T}$ is relatively small.

### 3.5.4 ESTIMATING $\hat{\Psi}$

The formulae for $\hat{F}^{-1}$ from the previous sections depend on the quantity $\hat{\Psi} = \hat{V}_1 = V_0^{-1/2}\Psi V_0^{1/2}$, and so it remains to compute $\Psi$. We observe that

$$V_1 = V_{1,0} = \mathbb{E}[w_1 w_0^{\top}] = \mathbb{E}[(\Psi w_0 + \epsilon_1)w_0^{\top}] = \Psi\mathbb{E}[w_0 w_0^{\top}] + \mathbb{E}[\epsilon_1 w_0^{\top}] = \Psi V_0 + 0 = \Psi V_0.$$

Right-multiplying both sides by $V_0$ yields $\Psi = V_1 V_0^{-1}$. Thus, given estimates of $V_0$ and $V_1$, we may compute an estimate of $\hat{\Psi}$ as

$$\hat{\Psi} = V_0^{-1/2}V_1 V_0^{-1/2}.$$

In practice we estimate $V_0$ and $V_1$ by forming estimates of their Kronecker factors and taking the product. The factors themselves are estimated using exponentially decayed averages over mini-batch estimates. And the mini-batch estimates are in turn computed by averaging over cases and summing across time-steps, before divide by the expected number of time-steps.

For example, for $A_0$ and $A_1$ these the mini-batch estimates are averages of $\frac{1}{\mathbb{E}_{\mathcal{T}}[\mathcal{T}]} \sum_{t=1}^{\mathcal{T}} a_t a_t^{\top}$ and $\frac{1}{\mathbb{E}_{\mathcal{T}}[\mathcal{T}]} \sum_{t=1}^{\mathcal{T}-1} a_{t+1} a_t^{\top}$, respectively. Note that as long as $V_0$ is computed as the 2nd-order moment of some empirical data, and $V_1$ computed as the 2nd-order moment between that same data and a temporally shifted version, the spectral radius of $\hat{\Psi} = V_0^{-1/2}V_1 V_0^{-1/2}$ (and similarly $\Psi = V_1 V_0^{-1}$) will indeed be less than or equal to 1, as we prove in Section B.2 of the appendix. This bound on the

spectral radius is a necessary condition for our infinite chain-structured Gaussian graphical model to be well-defined, and for our calculations to make sense.

The sufficient condition that the spectral radius is actually less than 1 will most often be satisfied too, except in the unlikely event that some eigen component remains *perfectly constant* across time. But even if this somehow happens, the inclusion within the given $V_0$ of some damping/regularization term such as $\lambda I$ will naturally deal with this problem.

### 3.5.5 EFFICIENT IMPLEMENTATION ASSUMING $V_0$ AND $V_1$ ARE KRONECKER-FACTORED

It remains to show that the approximations developed in Section 3.5 can be combined with the Kronecker-factored approximations for $V_0$ and $V_1$ from Section 3.2.3 to yield an efficient algorithm for computing $F^{-1}z$ for an arbitrary vector $z = \text{vec}(Z)$. This is a straightforward although very long computation which we leave to Section C of the appendix.

Full pseudo-code for the resulting algorithms is given in Section C.3. As they only involve symmetric eigen-decomposition and matrix-matrix products with matrices the size of $A_0$ and $G_0$ they are only several times more expensive to compute than eqn. 6. This extra overhead will often be negligible since the gradient computation via BPTT, whose costs scales with the sequence length $\mathcal{T}$, tends to dominate all the other costs.

## 4 EXPERIMENTS

To demonstrate the benefit of our novel curvature matrix approximations for RNNs, we empirically evaluated them within the standard "distributed K-FAC" framework (Ba et al., 2017) on two different RNN training tasks.

The 2nd-order statistics (i.e. the Kronecker factors $A_0$, $A_1$, $G_0$, and $G_1$) are accumulated through an exponential moving average during training. When computing our approximate inverse Fisher, factored Tikhonov damping (Martens & Grosse, 2015) was applied to $V_0 = G_0 \otimes A_0$.

We used a single machine with 16 CPU cores and a Nvidia K40 GPU for all the experiments. The additional computations required to get the approximate Fisher inverse from these statistics (i.e. the "pre-processing steps" described in Section C.3) are performed asynchronously on the CPUs, while the GPU is used for the usual forward evaluation and back-propagation to compute the gradient. Updates are computed using the most recently computed values of these (which are allowed to be stale), so there is minimal per-iteration computational overhead compared SGD.

We adopted the step-size selection technique described in Section 5 of Ba et al. (2017), as we found it let us use larger learning rates without compromising the stability of the optimization. The hyper-parameters of our approach, which include the max learning rate and trust-region size for the afore-mentioned step-size selection procedure, as well as the momentum, damping constants, and the decay-rate for the second-order statistics, as well as the hyper-parameters of the baseline methods, were tuned using a grid search.

**Word-level language model**: We start by applying our method to a two-layer RNN based on the well-studied Long Short-Term Memory (LSTM) architecture (Hochreiter & Schmidhuber, 1997) for a word-level language modeling task on the Penn-TreeBank (PTB) dataset (Marcus et al., 1993) following the experimental setup in Zaremba et al. (2014). The gradients are computed using a fixed sequence length truncated back-propagation scheme in which the initial states of the recurrent hidden units are inherited from the final state of the preceding sequence. The truncation length used in the experiments is 35 timesteps. The learning rate is given by a carefully tuned decaying schedule (whose base value we tune along with the other hyperparamters).

In our experiments we simply substitute their optimizer with our modified distributed K-FAC optimizer that uses our proposed RNN Fisher approximations. We performed experiments on two different sizes of the same architecture, which use two-layer 650 and 1024 LSTM units respectively.

LSTMs have 4 groups of internal units: input gates, output gates, forget gates, and update candidates. We treat the 4 weight matrices that compute the pre-activations to each of these as distinct for the purposes of defining Fisher blocks (whereas many LSTM implementations treat them as one big matrix). This results in smaller Kronecker factors that are cheaper to compute and invert.

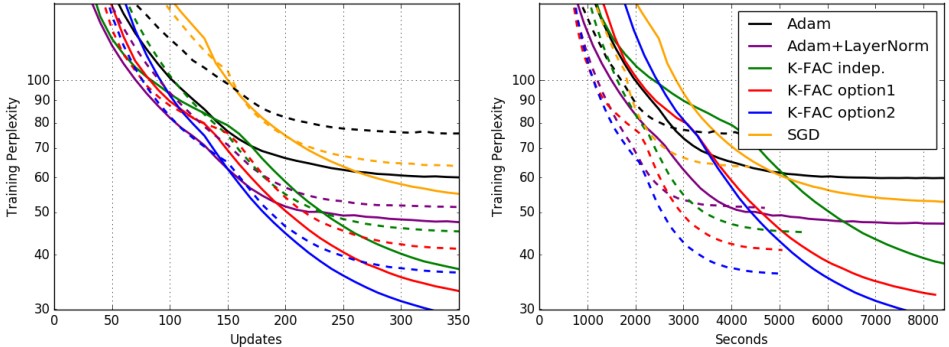

Figure 1: Optimization performance of our method compared to the baselines in perplexity-per-word on length-35 word sequences from Penn-TreeBank. All the methods used a mini-batch size of 200. *K-FAC indep.* uses the update in eqn. 6, *K-FAC option1* uses eqn. 8, and *K-FAC option2* uses eqn. 9. (left) Training perplexity v.s. the number of updates. Dashed lines denote the training curves for RNNs with 1024 LSTM units and solid lines denote the training curves for RNNs with 650 LSTM units. (right) Training perplexity v.s. the wall-clock time.

## 4.1 LANGUAGE MODELING WITH LONG SHORT-TERM MEMORY UNITS

Because the typical vocabulary size used for PTB is 10,000, the Fisher blocks for the input embedding layer and output layer (computing the logits to the softmax) each contain a 10,000 by 10,000 sizes Kronecker factor, which is too large to be inverted with any reasonable frequency. Given that the input vector uses a one-hot encoding it is easy to see its associated factor is actually diagonal, and so we can store and invert it as such. Meanwhile the large factor associated with the output isn't diagonal, but we nonetheless approximate it as such for the sake of efficiency.

In our experiments we found that each parameter update of our method required about 80% more wall-clock time than an SGD update (using mini-batch size of 200) although the updates made more much progress.

In Figure 1, we plot the training progress as a function of the number of parameter updates. While Adam outperforms SGD in the first few epochs, SGD obtains a lower loss at the end of training. We found the recent layer-normalization technique (Ba et al., 2016) helps speed up Adam considerably, but it hurts the SGD performance. Such an observation is consistent with previous findings. In comparison, our proposed method still significantly outperform both the Adam and the SGD baselines even with the help of layer-normalization.

While optimization performance, not generalization performance, is the focus of this paper, we have included validation performance data in the appendix for the sake of completeness. (See Figure 4 in Appendix D.) Not surprisingly, we found that the 2nd-order methods, including our approach and diagonal ones like Adam, tended to overfit more than SGD on these tasks.

The tendency for SGD w/ early-stopping to self-regularize is well-documented, and there are many compelling theories about why this happens (e.g. Duvenaud et al., 2016; Hardt et al., 2015). It is also well-known that 2nd-order methods, including K-FAC and diagonal methods like Adam/RMSprop, dont self-regularize nearly as much (e.g. Wilson et al., 2017; Keskar & Socher, 2017). We feel that this problem can likely be addressed through the careful application of additional explicit regularization (e.g. increased weight decay, drop-out, etc) and/or model modifications, but that exploring this is outside of the scope of this paper.

**Character-level model**: To further investigate the optimization performance of our proposed Fisher approximation, we use a small two layer LSTM with 128 units to model the character sequences on the Penn-TreeBank (PTB) dataset (Marcus et al., 1993). We employ the same data partition in Mikolov et al. (2012). We plotted the bits-per-character vs the number of parameter updates and the wall-clock times in Figure 2. The K-FAC updates were roughly twice as time-consuming to compute as the Adam updates in our implementation. Despite this, our results demonstrate that K-FAC has a significant advantage over the Adam baseline in terms of wall-clock time.

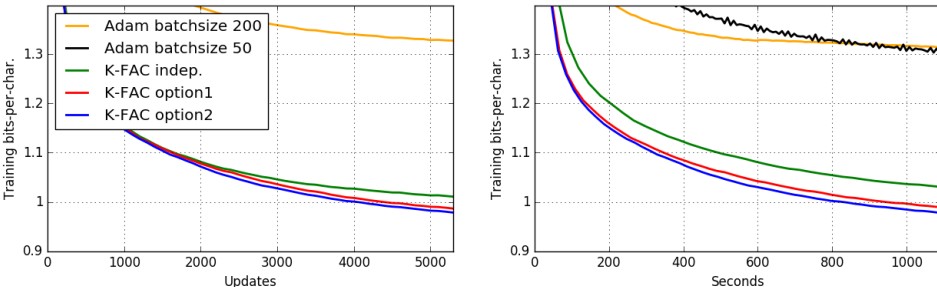

Figure 2: Optimization performance in bit-per-character on length-100 character sequences from Penn-TreeBank. *batchsize* indicates the mini-batch size used to train the baseline methods (our method always used a mini-batch size of 200). *K-FAC indep.* uses the update in eqn. 6, *K-FAC option1* uses eqn. 8, and *K-FAC option2* uses eqn. 9. (left) Training perplexity v.s. the number of updates. (right) Training perplexity v.s. the wall-clock time.

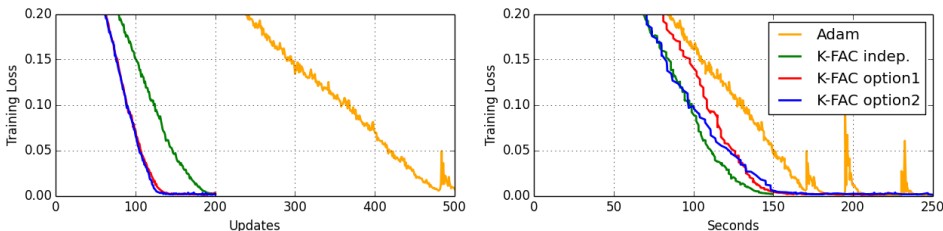

Figure 3: Optimization performance for differentiable Neural Computers (DNC) on a repeated copy task. *K-FAC indep.* uses the update in eqn. 6, *K-FAC option1* uses eqn. 8, and *K-FAC option2* uses eqn. 9. (left) Training cross entropy loss v.s. the number of updates. (right) Training cross entropy loss v.s. the wall-clock time.

## 4.2 LEARNING DIFFERENTIABLE NEURAL COMPUTERS

To further investigate the potential benefits of using our approach over existing methods, we applied it to the Differentiable Neural Computer (DNC) model (Graves et al., 2016) for learning simple algorithmic programs. Recently, there have been several attempts (Weston et al., 2014; Graves et al., 2016) to extend the existing RNN models to incorporate more long-term memory storage devices in order to help solve problems beyond simple sequence prediction tasks. Although these extended RNNs could potentially be more powerful than simple LSTMs, they often require thousands of parameter updates to learn simple copy tasks (Graves et al., 2016). Both the complexity of these models and the difficulty of the learning tasks have posed a significant challenge to commonly used optimization methods.

The DNC model is designed to solve structured algorithmic tasks by using an LSTM to control an external read-write memory. We applied the Fisher-based precondition to compute the updates for both the weights in the LSTM controller and the read-write weight matrices used to interface with the memory. We trained the model on a simple repeated copy task in which the DNC needs to recreate a series of two random binary sequences after they are presented as inputs. The total length of the sequence is fixed to 22 time-steps. From Figure 3, we see that our method significantly outperforms the Adam baseline in terms of update count, although only provides a modest improvement in wall-clock time.

This gap is explained by the fact that the iterations were significantly more time-consuming to compute relative to the gradient computations than they were in previous two experiments on language models. This is likely due to a different trade-off in terms of the gradient computation vs the overheads specific to our method owing to smallness of the model and dataset. With more careful engineering to reduce the communication costs, and/or a larger model and dataset, we would expect to see a bigger improvement in wall-clock time.

## 5 CONCLUSION

We have presented a new family of approximations to the Fisher information matrix of recurrent neural networks (RNNs), extending previous work on Kronecker-factored approximations. With this contribution, recurrent networks can now finally be trained with the K-FAC optimization method. We have demonstrated that our new approximations substantially reduce the required number of iterations for convergence vs standard baseline optimizers on several realistic tasks. And we have also shown that in a modern distributed training setup this results in a substantial savings in wall-clock time as well.

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

# A    SUPPLEMENTARY COMPUTATIONS

## A.1    PROOFS FOR SECTION 3.5.1

**Proposition 1** *Given*

$$\hat{F}_{\mathcal{T}} = V_0^{1/2} F_{\mathcal{T}} V_0^{1/2} \quad and \quad \hat{\Psi} = \hat{V}_1 = V_0^{-1/2} \Psi V_0^{1/2}.$$

*we have*

$$\hat{F}_{\mathcal{T}} = \zeta_{\mathcal{T}}(\hat{\Psi}) + \zeta_{\mathcal{T}}(\hat{\Psi}^{\top}) - \mathcal{T}I$$

*where*

$$\zeta_{\mathcal{T}}(x) = \frac{\mathcal{T}(1-x) - x(1-x^{\mathcal{T}})}{(1-x)^2}.$$

**Proof**

For $d > 0$ we have that $V_d = \Psi V_{d-1}$, which can be seen as follows:

$$
\begin{aligned}
V_d &= V_{d,0} = \mathbb{E}[w_d w_0^{\top}] = \mathbb{E}[\Psi w_{d-1} + \epsilon_d w_0^{\top}] \\
&= \Psi \mathbb{E}[w_{d-1} w_0^{\top}] + \mathbb{E}[\epsilon_d w_0^{\top}] \\
&= \Psi V_{d-1} + 0 = \Psi V_{d-1}.
\end{aligned}
$$

Applying $V_d = \Psi V_{d-1}$ recursively yields

$$V_d = \Psi^d V_0 \quad \text{for} \quad d \geqslant 0.$$

And using $V_{-d} = V_d^{\top}$ it also follows that

$$V_d = V_0 (\Psi^d)^{\top} \quad \text{for} \quad d \leqslant 0.$$

Setting $d = 1$ and multiplying both sides by $V_0$ (which is assumed to be invertible) one can also derive the following simple formula for $\Psi$:

$$\Psi = V_1 V_0^{-1}. \tag{10}$$

To proceed from here we define a "transformed" version of the original chain-structured linear-Gaussian graphical model whose variables are $\hat{w}_t = V_0^{-1/2} w_t$. (Here we assume that $V_0$ is invertible – it is symmetric by definition.) All quantities related to the original model have their analogues in the transformed model, which we indicate with the hat symbol $\hat{\cdot}$.

In the transformed model the 2nd-order moments of the $\hat{w}_t$'s are given by

$$\hat{V}_d = \mathbb{E}\left[ \left( V_0^{-1/2} w_d \right) \left( V_0^{-1/2} w_0 \right)^{\top} \right] = V_0^{-1/2} \mathbb{E}\left[ w_d w_0^{\top} \right] V_0^{-1/2} = V_0^{-1/2} V_d V_0^{-1/2}.$$

We observe that $\hat{V}_0 = I$.

Analogously to the original model, the transformed version obeys

$$\hat{w}_t = \hat{\Psi} \hat{w}_{t-1} + \hat{\epsilon}_t,$$

with $\hat{\epsilon}_t = V_0^{-1/2} \epsilon_t$ and $\hat{\Psi} = \hat{V}_1 \hat{V}_0^{-1} = \hat{V}_1$ (using $\hat{V}_0 = I$). This can be seen by noting that

$$
\begin{aligned}
\hat{w}_t &= V_0^{-1/2} w_t \\
&= V_0^{-1/2} (\Psi w_{t-1} + \epsilon_t) \\
&= V_0^{-1/2} V_1 V_0^{-1} w_{t-1} + V_0^{-1/2} \epsilon_t \\
&= (V_0^{-1/2} V_1 V_0^{-1/2})(V_0^{-1/2} w_{t-1}) + \hat{\epsilon}_t \\
&= \hat{V}_1 \hat{w}_{t-1} + \hat{\epsilon}_t.
\end{aligned}
$$

It also remains true that the spectral radius of $\hat{\Psi}$ is less than 1, which can be seen in at least one of two ways: by noticing that the transformed model is well-defined in the infinite limit if and only if

the original one is, or that $\hat{\Psi} = \hat{V}_1 = V_0^{-1/2}\Psi V_0^{1/2}$ is a similar matrix to $\Psi$ (in the technical sense) and hence has the same eigenvalues.

As the transformed model is isomorphic to the original one, all of the previously derived relationships which held for it also hold here, simply by replacing each quantity with its transformed version (denoted by the hat symbol $\hat{\cdot}$).

Given these relations (included the transformed analogue of equation 5) we can express $\hat{F}_{\mathcal{T}}$ as

$$\hat{F}_{\mathcal{T}} = \sum_{d=0}^{\mathcal{T}}(\mathcal{T}-d)\hat{\Psi}^d + \left(\sum_{d=0}^{\mathcal{T}}(\mathcal{T}-d)\hat{\Psi}^d\right)^{\top} - \mathcal{T}I.$$

It is a well-known fact that one can evaluate rational functions, and functions that are the limiting values of sequences of rational functions, with matrix arguments. This is done by replacing scalar multiplication with matrix multiplication, division with matrix inversion, and scalar constants with scalar multiples of the identity matrix, etc. Note that because sums of powers and inverses of matrices are co-diagonalizable/commutative when the matrices themselves are, there is no issue of ambiguity caused by mixing commutative and non-commutative algebra in this way.

Moreover, the value of some such function $f(x)$, given a matrix argument $B$, is

$$f(B) = V\,\mathrm{diag}(f(b))V^{-1},$$

where $V\,\mathrm{diag}(b)V^{-1} = B$ is eigendecomposition of B, and where $f(b)$ denotes the component-wise evaluation of $f$ for each component of the vector $b$, i.e. $[f(b)]_i = f([b]_i)$. Note that if $[f(b)]_i$ is undefined from some $i$, either because of a division by zero, or because the limit which defines $f(x)$ doesn't converge for $x = [b]_i$, then $f(B)$ doesn't exist for that particular $B$ (and otherwise it does).

We observe that our above expression for $\widehat{F}_{\mathcal{T}}$ can be rewritten as

$$\hat{F}_{\mathcal{T}} = \zeta_{\mathcal{T}}(\hat{\Psi}) + \zeta_{\mathcal{T}}(\hat{\Psi}^{\top}) - \mathcal{T}I,$$

where $\zeta_{\mathcal{T}}(x) = \sum_{d=0}^{\mathcal{T}}(\mathcal{T}-d)x^d$. By Proposition 3 in Appendix B.1, we have for $x \neq 1$ that

$$\zeta_{\mathcal{T}}(x) = \frac{\mathcal{T}(1-x) - x(1-x^{\mathcal{T}})}{(1-x)^2}.$$

Let $U\,\mathrm{diag}(\hat{\psi})U^{-1} = \hat{\Psi}$ be the eigendecomposition of $\hat{\Psi}$. Because $\hat{\Psi}$ has a spectral radius less than 1, we have $|[\hat{\psi}]_i| < 1$ for each $i$ (so that in particular $[\hat{\psi}]_i \neq 1$), and thus we can evaluate $\zeta_{\mathcal{T}}(\hat{\Psi})$ and $\zeta_{\mathcal{T}}(\hat{\Psi}^{\top})$ according to the above formula for $\zeta_{\mathcal{T}}(x)$.

$\square$

## A.2 Proofs for Section 3.5.3

**Proposition 2** *Suppose we approximate* $\hat{F} = \mathbb{E}_{\mathcal{T}}[\hat{F}_{\mathcal{T}}^{(\infty)}]$, *where we have defined*

$$\hat{F}_{\mathcal{T}}^{(\infty)} \equiv \lim_{\mathcal{T}'\to\infty}\frac{\mathcal{T}}{\mathcal{T}'}\hat{F}_{\mathcal{T}'}.$$

*Then we have*

$$\hat{F}^{-1} = \frac{1}{\mathbb{E}_{\mathcal{T}}[\mathcal{T}]}(I - \hat{\Psi})(I - \hat{\Psi}^{\top}\hat{\Psi})^{-1}(I - \hat{\Psi}^{\top}).$$

**Proof**

From eqn. 7 we have that

$$\begin{aligned}
\hat{F}_{\mathcal{T}}^{(\infty)} &= \lim_{\mathcal{T}'\to\infty}\frac{\mathcal{T}}{\mathcal{T}'}(\zeta_{\mathcal{T}'}(\hat{\Psi}) + \zeta_{\mathcal{T}'}(\hat{\Psi}^{\top}) - \mathcal{T}'I) \\
&= \lim_{\mathcal{T}'\to\infty}\frac{\mathcal{T}}{\mathcal{T}'}\zeta_{\mathcal{T}'}(\hat{\Psi}) + \lim_{\mathcal{T}'\to\infty}\frac{\mathcal{T}}{\mathcal{T}'}\zeta_{\mathcal{T}'}(\hat{\Psi}^{\top}) - \lim_{\mathcal{T}'\to\infty}\frac{\mathcal{T}}{\mathcal{T}'}\mathcal{T}'I.
\end{aligned}$$

To evaluate this we first term note that

$$\lim_{\mathcal{T}' \to \infty} \frac{\mathcal{T}}{\mathcal{T}'} \mathcal{T}' I = \mathcal{T} I \quad \text{and} \quad \lim_{\mathcal{T}' \to \infty} \frac{\mathcal{T}}{\mathcal{T}'} \zeta_{\mathcal{T}'}(A) = \kappa(A),$$

where we have defined

$$\kappa(x) = \lim_{\mathcal{T}' \to \infty} \frac{\mathcal{T}}{\mathcal{T}'} \zeta_{\mathcal{T}'}(x).$$

For $|x| < 1$ we have that $\lim_{\mathcal{T}' \to \infty} x^{\mathcal{T}'} = 0$, from which it follows that

$$
\begin{aligned}
\kappa(x) &= \lim_{\mathcal{T}' \to \infty} \frac{\mathcal{T}}{\mathcal{T}'} \left( \frac{\mathcal{T}'(1-x) - x(1-x^{\mathcal{T}'})}{(1-x)^2} \right) \\
&= \lim_{\mathcal{T}' \to \infty} \frac{\mathcal{T}}{\mathcal{T}'} \frac{\mathcal{T}'}{1-x} - \lim_{\mathcal{T}' \to \infty} \frac{\mathcal{T}}{\mathcal{T}'} \frac{x(1-x^{\mathcal{T}'})}{(1-x)^2} \\
&= \lim_{\mathcal{T}' \to \infty} \frac{\mathcal{T}}{1-x} - \lim_{\mathcal{T}' \to \infty} \frac{\mathcal{T}}{\mathcal{T}'} \lim_{\mathcal{T}' \to \infty} \frac{x(1-x^{\mathcal{T}'})}{(1-x)^2} \\
&= \frac{\mathcal{T}}{1-x} - \lim_{\mathcal{T}' \to \infty} \frac{\mathcal{T}}{\mathcal{T}'} \frac{x(1-0)}{(1-x)^2} \\
&= \frac{\mathcal{T}}{1-x}.
\end{aligned}
$$

Let $U \operatorname{diag}(\hat{\psi}) U^{-1} = \hat{\Psi}$ be the eigendecomposition of $\hat{\Psi}$. Using the fact that $|[\hat{\psi}]_i| < 1$ (as established in Section A.1) we can use the above expression to evaluate $\kappa(x)$ at both $x = \hat{\Psi}$ and $x = \hat{\Psi}^\top$, which yields

$$\hat{F}_{\mathcal{T}}^{(\infty)} = \kappa(\hat{\Psi}) + \kappa(\hat{\Psi}^\top) = \mathcal{T}((I - \hat{\Psi})^{-1} + (I - \hat{\Psi}^\top)^{-1} - I).$$

Pre-multiplying both sides by $I - \hat{\Psi}^\top$, and post-multiplying both sides by $I - \hat{\Psi}$, we have

$$
\begin{aligned}
(I - \hat{\Psi}^\top) \hat{F}_{\mathcal{T}}^{(\infty)} (I - \hat{\Psi}) &= \mathcal{T}((I - \hat{\Psi}^\top) + (I - \hat{\Psi}) - (I - \hat{\Psi}^\top)(I - \hat{\Psi})) \\
&= \mathcal{T}(I - \hat{\Psi}^\top + I - \hat{\Psi} - I + \hat{\Psi}^\top + \hat{\Psi} - \hat{\Psi}^\top \hat{\Psi}) \\
&= \mathcal{T}(I - \hat{\Psi}^\top \hat{\Psi}).
\end{aligned}
$$

Then applying the reverse operation gives

$$\widehat{F}_{\mathcal{T}}^{(\infty)} = \mathcal{T}(I - \hat{\Psi}^\top)^{-1}(I - \hat{\Psi}^\top \hat{\Psi})(I - \hat{\Psi})^{-1}.$$

Taking the expectation over $\mathcal{T}$ gives

$$\hat{F} = \mathbb{E}_{\mathcal{T}}[\hat{F}_{\mathcal{T}}^{(\infty)}] = \mathbb{E}_{\mathcal{T}}[\mathcal{T}](I - \hat{\Psi}^\top)^{-1}(I - \hat{\Psi}^\top \hat{\Psi})(I - \hat{\Psi})^{-1}.$$

Finally, inverting both sides yields

$$\hat{F}^{-1} = \frac{1}{\mathbb{E}_{\mathcal{T}}[\mathcal{T}]}(I - \hat{\Psi})(I - \hat{\Psi}^\top \hat{\Psi})^{-1}(I - \hat{\Psi}^\top).$$

$\square$

# B ADDITIONAL TECHNICAL PROOFS

## B.1

**Proposition 3** *Suppose $x \in \mathbb{C}$, $x \neq 0$, and $\mathcal{T}$ is a non-negative integer. We have*

$$\sum_{i=0}^{\mathcal{T}} (\mathcal{T} - i)x^i = \frac{\mathcal{T}(1-x) - x(1-x^{\mathcal{T}})}{(1-x)^2}.$$

**Proof** Observe that

$$x \frac{\mathrm{d} \sum_{i=0}^{\mathcal{T}} x^i}{\mathrm{d}x} = x \sum_{i=0}^{\mathcal{T}} \frac{\mathrm{d}x^i}{\mathrm{d}x} = x \sum_{i=0}^{\mathcal{T}} ix^{i-1} = \sum_{i=0}^{\mathcal{T}} ix^i.$$

Another way to express this is to use the geometric series formula $\sum_{i=0}^{\mathcal{T}} x^i = \frac{1-x^{\mathcal{T}+1}}{1-x}$ (which holds for $x \neq 0$) before computing the derivative. This gives

$$x \frac{\mathrm{d} \sum_{i=0}^{\mathcal{T}} x^i}{\mathrm{d}x} = x \frac{\mathrm{d} \left( \frac{1-x^{\mathcal{T}+1}}{1-x} \right)}{\mathrm{d}x} = x \left( -\frac{(1+\mathcal{T})x^{\mathcal{T}}}{1-x} + \frac{1-x^{\mathcal{T}+1}}{(1-x)^2} \right) = \frac{x(1 - x^{\mathcal{T}+1} - (1+\mathcal{T})(1-x)x^{\mathcal{T}})}{(1-x)^2}.$$

Thus we have

$$\sum_{i=0}^{\mathcal{T}} ix^i = \frac{x(1 - x^{\mathcal{T}+1} - (1+\mathcal{T})(1-x)x^{\mathcal{T}})}{(1-x)^2}.$$

And so

$$
\begin{aligned}
\sum_{i=0}^{\mathcal{T}} (\mathcal{T} - i)x^i &= \mathcal{T} \sum_{i=0}^{\mathcal{T}} x^i - \sum_{i=0}^{\mathcal{T}} ix^i \\
&= \frac{\mathcal{T}(1 - x^{\mathcal{T}+1})}{1-x} - \frac{x(1 - x^{\mathcal{T}+1} - (1+\mathcal{T})(1-x)x^{\mathcal{T}})}{(1-x)^2} \\
&= \frac{\mathcal{T}(1-x)(1 - x^{\mathcal{T}+1}) - x(1 - x^{\mathcal{T}+1} - (1+\mathcal{T})(1-x)x^{\mathcal{T}})}{(1-x)^2} \\
&= \frac{\mathcal{T} - \mathcal{T}x - \mathcal{T}x^{\mathcal{T}+1} + \mathcal{T}x^{\mathcal{T}+2} - x + x^{\mathcal{T}+2} + x^{\mathcal{T}+1}(1 - x + \mathcal{T} - \mathcal{T}x)}{(1-x)^2} \\
&= \frac{\mathcal{T} - \mathcal{T}x - \mathcal{T}x^{\mathcal{T}+1} + \mathcal{T}x^{\mathcal{T}+2} - x + x^{\mathcal{T}+2} + x^{\mathcal{T}+1} - x^{\mathcal{T}+2} + \mathcal{T}x^{\mathcal{T}+1} - \mathcal{T}x^{\mathcal{T}+2}}{(1-x)^2} \\
&= \frac{\mathcal{T} - (\mathcal{T}+1)x + x^{\mathcal{T}+1}}{(1-x)^2} \\
&= \frac{\mathcal{T}(1-x) - x(1 - x^{\mathcal{T}})}{(1-x)^2}
\end{aligned}
$$

where we have again used the geometric series formula $\sum_{i=0}^{\mathcal{T}} x^i = \frac{1-x^{\mathcal{T}+1}}{1-x}$ on the second line. $\square$

## B.2 SPECTRAL BOUND FOR ESTIMATE OF $\hat{\Psi}$

In what follows all quantities are computed using their defining formulae, starting from the estimated values of $A_0$, $A_1$, $G_0$, and $G_1$.

First we observe that since $\hat{\Psi} = \hat{V}_1 = V_0^{-1/2} \Psi V_0^{1/2}$ is similar to $\Psi$ (in the technical sense of the word), they share the same eigenvalues. Thus it suffices bound to the spectral radius of $\Psi = V_1 V_0^{-1}$.

Next we observe that $V_0 = A_0 \otimes G_0$ and $V_1 = A_1 \otimes G_1$, so that

$$V_1 V_0^{-1} = (A_1 \otimes G_1)(A_0 \otimes G_0)^{-1} = (A_1 A_0^{-1}) \otimes (G_1 G_0^{-1}).$$

Because the eigendecomposition of a Kronecker product is the Kronecker product of the decompositions of the factors we have that $\rho(V_1 V_0^{-1}) = \rho(A_1 A_0^{-1})\rho(G_1 G_0^{-1})$, where $\rho(X)$ denotes the spectral radius of a matrix $X$.

Thus it suffices to show that $\rho(A_1 A_0^{-1}) \leq 1$ and $\rho(G_1 G_0^{-1}) \leq 1$ for $A_i$ and $G_i$ as computed by the estimation scheme outlined in Section 3.5.4. Recall that this is the exponentially decayed average of mini-batch averages of estimators of the form $\frac{1}{\mathbb{E}_{\mathcal{T}}[\mathcal{T}]} \sum_{t=1}^{\mathcal{T}} a_t a_t^{\top}$ and $\frac{1}{\mathbb{E}_{\mathcal{T}}[\mathcal{T}]} \sum_{t=1}^{\mathcal{T}-1} a_{t+1} a_t^{\top}$.

In the remainder of this section we will show that $\rho(A_1 A_0^{-1}) \leq 1$. The argument to show that $\rho(G_1 G_0^{-1}) \leq 1$ is identical.

Define

$$M_0 = [0 \quad a_1 \quad a_2 \quad \cdots \quad a_{\mathcal{T}}]$$

and

$$M_1 = [a_1 \quad a_2 \quad \cdots \quad a_{\mathcal{T}} \quad 0]$$

We observe that $M_0 M_0^\top = M_1 M_1^\top = \sum_{t=1}^{\mathcal{T}} a_t a_t^\top$ and $M_1 M_0^\top = \sum_{t=1}^{\mathcal{T}-1} a_{t+1} a_t^\top$.

Provided that the exponentially decayed averages for $A_0$ and $A_1$ are computed in the same way and use the same normalizers (i.e. $1/(m\mathbb{E}_{\mathcal{T}}[\mathcal{T}])$) we thus have that the matrix

$$\mathcal{A} = \begin{bmatrix} A_0 & A_1 \\ A_1^\top & A_0 \end{bmatrix}$$

is a positively-weighted linear combination of terms of the form

$$\begin{bmatrix} M_1 \\ M_0 \end{bmatrix} \begin{bmatrix} M_1 \\ M_0 \end{bmatrix}^\top \succeq 0$$

where the various $a_t$'s are computed on different data using current and previous model parameters. It thus follows that $\mathcal{A} \succeq 0$. The inequality $\rho(A_1 A_0^{-1}) \leq 1$ now follows from the following lemma.

**Lemma 1** *Consider a real, symmetric, positive semi-definite block matrix*

$$\begin{bmatrix} B & C \\ C^\top & B \end{bmatrix} \succeq 0. \tag{11}$$

*If $B$ is invertible then we have $\rho(CB^{-1}) \leq 1$, and further if the block matrix is positive definite we have $\rho(CB^{-1}) < 1$.*

**Proof** Because similarity transformations preserve eigenvalues, the statement is equivalent to $\rho(B^{-\frac{1}{2}}CB^{-\frac{1}{2}}) \leq 1$. Define $X \triangleq B^{-\frac{1}{2}}CB^{-\frac{1}{2}}$. Because any induced matrix norm is an upper-bound on the spectral radius, it suffices to show $\|X\|_2 = \sigma_{\max}(X) \leq 1$, where $\sigma_{\max}(X)$ denotes the largest singular value of $X$.

By taking Schur complements of the block matrix (11) we have

$$0 \preceq B - CB^{-1}C^\top = B^{\frac{1}{2}}(I - (B^{-\frac{1}{2}}CB^{-\frac{1}{2}})(B^{-\frac{1}{2}}CB^{-\frac{1}{2}}))B^{\frac{1}{2}}$$
$$= B^{\frac{1}{2}}(I - XX^\top)B^{\frac{1}{2}}.$$

Note that the Schur complement is PSD because the original block matrix itself is (e.g. Zhang, 2006).

Using the fact that $Z \mapsto B^{\frac{1}{2}}ZB^{\frac{1}{2}}$ maps positive semidefinite matrices to positive semidefinite matrices, this implies

$$0 \preceq I - XX^\top \Rightarrow XX^\top \preceq I \Rightarrow \|X\|_2^2 \leq 1 \Rightarrow \rho(X) \leq 1.$$

When the block matrix is positive definite, the inequalities become strict. $\quad\square$

## C  EFFICIENT IMPLEMENTATION ASSUMING $V_0$ AND $V_1$ ARE KRONECKER-FACTORED

The ultimate goal of our calculations is to efficiently compute the matrix-vector product of some arbitrary vector $z$ (which will often be the gradient $\text{vec}(\mathcal{D}W)$) with our inverse Fisher approximation $F^{-1}$. That is, we wish to compute $F^{-1}z$.

It will be convenient to assume that $z$ is given as a matrix $Z$ (with the same dimensions as $\mathcal{D}W$) so that $z = \text{vec}(Z)$.

Multiplying the vector $z$ by $F^{-1} = V_0^{-1/2}\hat{F}^{-1}V_0^{-1/2}$ amounts to first multiplying by $V_0^{-1/2}$, then by $\hat{F}^{-1}$, and then by $V_0^{-1/2}$ again.

We will suppose that we are given $A_0$, $A_1$, $G_0$, and $G_1$ such that

$$V_0 = A_0 \otimes G_0 \quad \text{and} \quad V_1 = A_1 \otimes G_1.$$

We then have

$$V_0^{-1/2} = A_0^{-1/2} \otimes G_0^{-1/2}$$

and thus to multiply by $V_0^{-1/2}$ we can use the identity $(C \otimes B) \operatorname{vec}(X) = \operatorname{vec}(BXC^\top)$, giving

$$V_0^{-1/2} z = \operatorname{vec}(G_0^{-1/2} Z A_0^{-1/2}).$$

In matrix form this is simply $G_0^{-1/2} Z A_0^{-1/2}$. Note that $A_0^{-1/2}$ and $G_0^{-1/2}$ can be computed using the eigendecompositions of $A_0$ and $G_0$, for example.

The procedure to efficiently multiply a vector $z$ by $\hat{F}^{-1}$ is more involved and depends on which approximation "option" we are using. However one immediate useful insight we can make before specializing to Option 1 or Option 2 is that $\hat{\Psi}$ can be written as a Kronecker product as follows:

$$
\begin{aligned}
\hat{\Psi} &= V_0^{-1/2} V_1 V_0^{-1/2} \\
&= (A_0 \otimes G_0)^{-1/2} (A_1 \otimes G_1)(A_0 \otimes G_0)^{-1/2} \\
&= (A_0^{-1/2} \otimes G_0^{-1/2})(A_1 \otimes G_1)(A_0^{-1/2} \otimes G_0^{-1/2}) \\
&= (A_0^{-1/2} A_1 A_0^{-1/2}) \otimes (G_0^{-1/2} G_1 G_0^{-1/2}) \\
&= \hat{\Psi}_A \otimes \hat{\Psi}_G.
\end{aligned}
$$

where we have defined $\hat{\Psi}_A = A_0^{-1/2} A_1 A_0^{-1/2}$ and $\hat{\Psi}_G = G_0^{-1/2} G_1 G_0^{-1/2}$.

### C.1 OPTION 1

For Option 1 we assume that $V_1$, and hence $\hat{\Psi} = \hat{V}_1 = V_0^{-1/2} V_1 V_0^{-1/2}$, is symmetric. We note that $V_1 = A_1 \otimes G_1$ will be symmetric if and only if both $A_1$ and $G_1$ are.

Our task is to compute the matrix-vector product

$$\hat{F}^{-1} z = U \operatorname{diag}(\gamma(\hat{\psi})) U^\top z$$

where $U \operatorname{diag}(\hat{\psi}) U^\top$ is the eigendecomposition of $\hat{\Psi}$, and $\gamma(x)$ is defined as in Section 3.5.3, eqn. 8. To do this we first multiply the vector by $U^\top$, then by $\operatorname{diag}(\gamma(\hat{\psi}))$, and then finally by $U$.

The eigendecomposition of $\hat{\Psi}$ can be computed efficiently using its Kronecker product structure. In particular, we compute the eigendecompositions of each factor as

$$U_A \operatorname{diag}(\hat{\psi}_A) U_A^\top = \hat{\Psi}_A \quad \text{and} \quad U_G \operatorname{diag}(\hat{\psi}_G) U_G^\top = \hat{\Psi}_G,$$

from which we can write the eigendecomposition $U \operatorname{diag}(\hat{\psi}) U^{-1}$ of $\hat{\Psi}$ as

$$
\begin{aligned}
\hat{\Psi} &= \hat{\Psi}_A \otimes \hat{\Psi}_G \\
&= (U_A \operatorname{diag}(\hat{\psi}_A) U_A^\top) \otimes (U_G \operatorname{diag}(\hat{\psi}_G) U_G^\top) \\
&= (U_A \otimes U_G)(\operatorname{diag}(\hat{\psi}_A) \otimes \operatorname{diag}(\hat{\psi}_G))(U_A^\top \otimes U_G^\top) \\
&= (U_A \otimes U_G) \operatorname{diag}(\operatorname{vec}(\hat{\psi}_G \hat{\psi}_A^\top))(U_A^\top \otimes U_G^\top).
\end{aligned}
$$

In other words, we have $U = U_A \otimes U_G$ and $\hat{\psi} = \operatorname{vec}(\hat{\psi}_G \hat{\psi}_A^\top)$.

To multiply $z$ by $U^\top = U_A^\top \otimes U_G^\top$ we use the identity $(C \otimes B) \operatorname{vec}(X) = \operatorname{vec}(BXC^\top)$ which gives

$$U^\top z = (U_A^\top \otimes U_G^\top) \operatorname{vec}(Z) = \operatorname{vec}(U_G^\top Z U_A).$$

Similarly, the multiplication of $z$ by $U = U_A \otimes U_G$ can be computed as $\operatorname{vec}(U_G Z U_A^\top)$.

Finally, multiplying $z$ by $\operatorname{diag}(\gamma(\hat{\psi})) = \operatorname{diag}(\gamma(\operatorname{vec}(\hat{\psi}_G \hat{\psi}_A^\top)))$ corresponds to entry-wise multiplication of $Z$ by a matrix $Y$, where $\operatorname{vec}(Y) = \gamma(\operatorname{vec}(\hat{\psi}_G \hat{\psi}_A^\top))$, or in other words $[Y]_{i,j} = \gamma([\hat{\psi}_G \hat{\psi}_A^\top]_{i,j}) = \gamma([\hat{\psi}_G]_i [\hat{\psi}_A]_j)$. Thus we have

$$\operatorname{diag}(\gamma(\hat{\psi})) z = \operatorname{diag}(\operatorname{vec}(Y)) \operatorname{vec}(Z) = \operatorname{vec}(Z \odot Y),$$

where $\odot$ denotes entry-wise multiplication of matrices.

In summary we have that $\hat{F}^{-1}z$ can be computed in matrix form as

$$U_G((U_G^\top Z U_A) \odot Y)U_A^\top$$

for $Y$ s.t. $[Y]_{i,j} = \gamma([\hat{\psi}_G]_i[\hat{\psi}_A]_j)$.

We note that computing $\gamma([\hat{\psi}_G]_i[\hat{\psi}_A]_j)$ is trivial since it is just a scalar evaluation of the rational function $\gamma(x)$.

## C.2 OPTION 2

For Option 2 we must compute the matrix-vector product

$$\hat{F}^{-1}z = \frac{1}{\sum_i \omega_i \mathcal{T}_i}(I - \hat{\Psi})(I - \hat{\Psi}^\top \hat{\Psi})^{-1}(I - \hat{\Psi}^\top)v.$$

To do this we will first multiply by $I - \hat{\Psi}^\top$, then by $(I - \hat{\Psi}^\top \hat{\Psi})^{-1}$, and then by $I - \hat{\Psi}$, before finally dividing the result by $\sum_i \omega_i \mathcal{T}_i$.

To compute the matrix-vector product $(I - \hat{\Psi}^\top)z$ we use the identity $(C \otimes B)\operatorname{vec}(X) = \operatorname{vec}(BXC^\top)$ while noting that $\hat{\Psi}^\top = (\hat{\Psi}_A \otimes \hat{\Psi}_G)^\top = \hat{\Psi}_A^\top \otimes \hat{\Psi}_G^\top$. This gives

$$\begin{aligned}
(I - \hat{\Psi}^\top)\operatorname{vec}(Z) &= \operatorname{vec}(Z) - (\hat{\Psi}_A^\top \otimes \hat{\Psi}_G^\top)\operatorname{vec}(Z) \\
&= \operatorname{vec}(Z) - \operatorname{vec}(\hat{\Psi}_G^\top Z \hat{\Psi}_A).
\end{aligned}$$

The matrix form of this is simply $Z - \hat{\Psi}_G^\top Z \hat{\Psi}_A$.

We may similarly compute $(I - \hat{\Psi})z$ in matrix form as $Z - \hat{\Psi}_G Z \hat{\Psi}_A^\top$.

The harder task is to compute $(I - \hat{\Psi}^\top \hat{\Psi})^{-1}z$, which is what we tackle next.

We first observe that

$$\begin{aligned}
I - \hat{\Psi}^\top \hat{\Psi} &= I \otimes I - (\hat{\Psi}_A^\top \otimes \hat{\Psi}_G^\top)(\hat{\Psi}_A \otimes \hat{\Psi}_G) \\
&= I \otimes I - (\hat{\Psi}_A^\top \hat{\Psi}_A) \otimes (\hat{\Psi}_G^\top \hat{\Psi}_G) \\
&\equiv I \otimes I - M_A \otimes M_G.
\end{aligned}$$

Given the eigendecompositions $E_A \operatorname{diag}(m_A)E_A^\top = M_A$ and $E_G \operatorname{diag}(m_G)E_G^\top = M_G$ we can thus compute the larger eigendecomposition as

$$\begin{aligned}
I \otimes I - M_A \otimes M_G &= (E_A \otimes E_G)(I \otimes I - \operatorname{diag}(m_A) \otimes \operatorname{diag}(m_G))(E_A^\top \otimes E_G^\top) \\
&= (E_A \otimes E_G)\operatorname{diag}(\mathbb{1} \otimes \mathbb{1} - m_A \otimes m_G)(E_A^\top \otimes E_G^\top) \\
&= (E_A \otimes E_G)\operatorname{diag}(\operatorname{vec}(\mathbb{1}\mathbb{1}^\top - m_G m_A^\top))(E_A^\top \otimes E_G^\top),
\end{aligned}$$

where $\mathbb{1}$ is the vector of ones.

Using the eigendecomposition the inverse can then be easily computed as

$$(I \otimes I - M_A \otimes M_G)^{-1} = (E_A \otimes E_G)\operatorname{diag}(\operatorname{vec}(\mathbb{1}\mathbb{1}^\top - m_G m_A^\top))^{-1}(E_A^\top \otimes E_G^\top).$$

Thus $(I - \hat{\Psi}^\top \hat{\Psi})^{-1}z$ may be computed by first multiplying by $(E_A^\top \otimes E_G^\top)$, then by $\operatorname{diag}(\operatorname{vec}(\mathbb{1}\mathbb{1}^\top - m_G m_A^\top))^{-1}$ (which in matrix form corresponds to element-wise division by $\mathbb{1}\mathbb{1}^\top - m_G m_A^\top$), and then by $E_A \otimes E_G$. The matrix form of this is

$$E_G((E_G^\top Z E_A) \oslash (\mathbb{1}\mathbb{1}^\top - m_G m_A^\top))E_A^\top,$$

where $B \oslash C$ denotes element-wise division of the matrix $B$ by the matrix $C$.

### C.3 Pseudo-code for the computation of $F^{-1}z$

Given $A_0$, $A_1$, $G_0$, and $G_1$ such that $V_0 = A_0 \otimes G_0$ and $V_1 = A_1 \otimes G_1$, the procedure to compute $F^{-1}z$ for an arbitrary vector $z = \text{vec}(Z)$ is as follows.

Pre-processing for Option 1 only:

- Ensure that $A_1$ and $G_1$ are exactly symmetric, and if not, symmertrize them via:

$$A_1 \leftarrow \frac{A_1 + A_1^\top}{2} \quad \text{and} \quad G_1 \leftarrow \frac{G_1 + G_1^\top}{2}$$

Pre-processing steps (both options):

- Compute the matrix square roots of the factors of $V_0$ (e.g. using eigendecompositions):

$$A_0^{-1/2} \quad \text{and} \quad G_0^{-1/2}$$

- Compute the factors of the transformed transition matrix $\hat{\Psi}$:

$$\hat{\Psi}_A = A_0^{-1/2} A_1 A_0^{-1/2} \quad \text{and} \quad \hat{\Psi}_G = G_0^{-1/2} G_1 G_0^{-1/2}$$

Pre-processing for Option 1 only:

- Compute the eigendecompositions of the factors of $\hat{\Psi}$

$$U_A \, \text{diag}(\hat{\psi}_A) U_A^\top = \hat{\Psi}_A \quad \text{and} \quad U_G \, \text{diag}(\hat{\psi}_G) U_G^\top = \hat{\Psi}_G$$

- Sum across the temporal correlations and then invert by performing the element-wise computation in eigenspace:

$$[Y]_{i,j} = \gamma([\hat{\psi}_G]_i [\hat{\psi}_A]_j) \quad \text{where} \quad \gamma(x) = \frac{(1-x)^2}{\sum_i \omega_i (\mathcal{T}_i(1-x^2) - 2x(1-x^{\mathcal{T}_i}))}$$

Pre-processing for Option 2 only:

- Compute the eigendecompositions of the factors of $\hat{\Psi}^\top \hat{\Psi}$ (or equivalently the SVD's of $\hat{\Psi}_A$ and $\hat{\Psi}_G$):

$$E_A \, \text{diag}(m_A) E_A^\top = \hat{\Psi}_A^\top \hat{\Psi}_A \quad \text{and} \quad E_G \, \text{diag}(m_G) E_G^\top = \hat{\Psi}_G^\top \hat{\Psi}_G$$

$z$-dependent calculations for Option 1:

- Multiply the input vector $z = \text{vec}(Z)$ by $V_0^{-1/2}$:

$$Z_0 = G_0^{-1/2} Z A_0^{-1/2}$$

- Multiply by $\hat{F}^{-1}$ using its eigendecomposition:

$$Z_1 = U_G((U_G^\top Z_0 U_A) \odot Y) U_A^\top$$

where $B \odot C$ denotes the element-wise product between matrices $B$ and $C$.

- Multiply the result by $V_0^{-1/2}$ and express as a vector:

$$Z_2 = G_0^{-1/2} Z_1 A_0^{-1/2} \quad \text{and} \quad F^{-1}z = \text{vec}(Z_2)$$

$z$-dependent calculations for Option 2:

- Multiply the input vector $z = \text{vec}(Z)$ by $V_0^{-1/2}$:

$$Z_0 = G_0^{-1/2} Z A_0^{-1/2}$$

- Multiply the result by $I - \hat{\Psi}^{\top}$:

$$Z_1 = Z_0 - \hat{\Psi}_G^{\top} Z_0 \hat{\Psi}_A$$

- Multiply the previous result by $(I - \hat{\Psi}^{\top}\hat{\Psi})^{-1}$ using its eigendecomposition:

$$Z_2 = E_G((E_G^{\top} Z_1 E_A) \oslash (\mathbb{1}\mathbb{1}^{\top} - m_G m_A^{\top})) E_A^{\top}$$

where $X \oslash Y$ denotes the element-wise division of $X$ by $Y$.

- Multiply the result by $I - \hat{\Psi}$

$$Z_3 = Z_2 - \hat{\Psi}_G Z_2 \hat{\Psi}_A^{\top}$$

- Normalize the result by $\mathbb{E}_{\mathcal{T}}[\mathcal{T}] = \sum_i \omega_i \mathcal{T}_i$ (for $\mathcal{T}_i$ and $\omega_i$ as defined at the bottom of Section 3.4):

$$Z_4 = \frac{1}{\sum_i \omega_i \mathcal{T}_i} Z_3$$

- Multiply the result by $V_0^{-1/2}$ and express as a vector:

$$Z_5 = G_0^{-1/2} Z_4 A_0^{-1/2} \quad \text{and} \quad F^{-1}z = \text{vec}(Z_5)$$

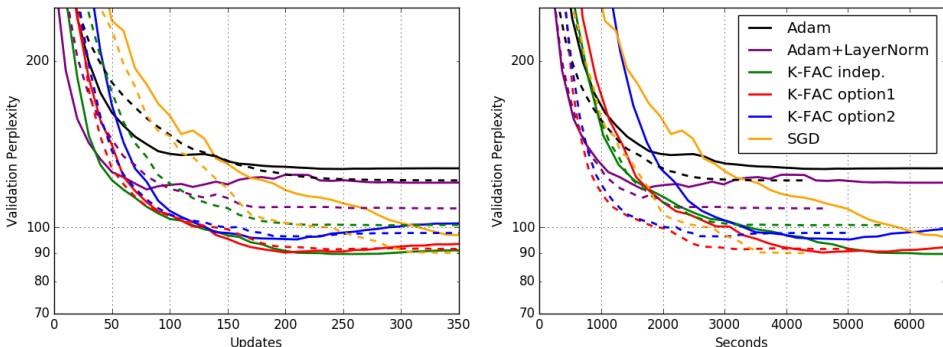

Figure 4: Generalization performance of our method compared to the baselines in perplexity-per-word on length-35 word sequences from Penn-TreeBank. All the methods used a mini-batch size of 200. *K-FAC indep.* uses the update in eqn. 6, *K-FAC option1* uses eqn. 8, and *K-FAC option2* uses eqn. 9. (left) Validation perplexity v.s. the number of updates. The dashed lines correspond to the experiments that used RNNs with 1024 LSTM units, and the solid lines correspond to experiments that used RNNs with 650 LSTM units. (right) Validation perplexity v.s. the wall-clock time.

# D    TEST PERFORMANCE ON PENN-TREEBANK

