# OpenReview forum: "Kronecker-factored Curvature Approximations for Recurrent Neural Networks"
_ICLR.cc/2018/Conference — Accept (Poster)_

### Official Review · AnonReviewer3 · 2017-11-27
**Assumptions used in approximations are not well justified**

**Rating:** 5
**Confidence:** 4

**Review:**

This paper extends the Kronecker-factor Approximate Curvature (K-FAC) optimization method to the setting of recurrent neural networks. The K-FAC method is an approximate 2nd-order optimization method that builds a block diagonal approximation of the Fisher information matrix, where the block diagonal elements are Kronecker products of smaller matrices.

In order to approximate the Fisher information matrix for RNNs, the authors assume that the derivative of the loss function with respect to each weight matrix at each time step is independent of the length of the sequence, that these derivatives are temporally homogeneous, that the input and derivatives of the output are independent across every point in time, and that either the one-step cross-covariance of these derivatives is symmetric or that the training sequences are effectively infinite in length. Based on these assumptions, the authors show that the Fisher information can be reduced into a form in which the derivatives of the weight matrices can be approximated by a linear Gaussian graphical model and in which the approximate 2nd order method can be efficiently carried out. The authors compare their method to SGD on two language modeling tasks and against Adam for learning differentiable neural computers.

The paper is relatively clear, and the authors do a reasonable job of introducing related work of the original K-FAC algorithm as well as its extension to CNNs before systematically deriving their method for RNNs. The problem of extending the K-FAC algorithm is natural, and the steps taken in this paper seem natural yet also original and non-trivial.

The main issue that I have with this paper is the lack of theoretical justification or even intuition for the many approximations carried out in the course of approximating the Fisher information matrix. In many instances, it seemed like these approximations were made purely for convenience and tractability without much regard for (even approximate) correctness. This quality of this paper would be greatly  strengthened if it had some bounds on approximation error or even empirical results testing the validity of the assumptions in the paper. Moreover, the experiments do not demonstrate levels of statistical significance in the results, so it is difficult to assert the practical significance of this work.

Specific comments and questions
Page 2, "r is is". Typo.
Page 2, "DV". I found the introduction of V without any explanation to be confusing.
Page 2, "P_{y|x}(\theta)". The relation between P_{y|x}(\theta) and f(x,\theta) is never explained.
Page 3, "common practice of computing the natural gradient as (F + \lambda I) \nabla h instead of F^{-1} \nabla h". I don't see how the former can serve as a replacement for the latter.
Page 3, "approximate g and a as statistically independent". Even though K-FAC already exists, it would be good to explain why this assumption is reasonable, since similar assumptions are made for the work presented in this paper.
Page 4, "This new approximation, called "KFC", is derived by assuming....". Same as previous comment. It would be good to briefly discuss why these assumptions are reasonable.
Page 5, Independence of T and w_t's, temporal homogeneity of w_t's,, and independence between a_t's and g_t's. I can see why these are convenient assumptions, but why are they reasonable? Moreover, why is it further natural to assume that A and G are temporally homogeneous as well?
Page 7, "But insofar as the w_t's ... encode the relevant information contained in these external variables, they should be approximately Markovian". I am not sure what this means.
Page 7, "The linear-Gaussian assumption meanwhile is a more severe one to make, but it seems necessary for there to be any hope that the required expectations remain tractable". I am not sure that this is a good enough justification for such an idea, unless there are compelling approximation error bounds.
Page 8, Option 1. In what situations is it reasonable to assume that V_1 is symmetric?
Pages 8-9, Option 2. What is a good finite sample size in which the assumption that the training sequences are infinitely long is reasonable in practice? Can the error |\kappa(x) - \zeta_T(x)| be translated into a statement on the approximation error?
Page 9, "V_1 = V_{1,0} = ...". Typos (that appear to have been caught by the authors already).
Page 9, "The 2nd-order statistics ... are accumulated through an exponential moving average during training". How sensitive is the performance of this method to the decay rate of the exponential moving average?
Page 10, "The additional computations required to get the approximate Fisher inverse from these statistics ... are performed asynchronously on the CPU's". I find it a bit unfair to compare SGD to K-FAC in terms of wall clock time without also using the extra CPU's for SGD as well (e.g. via Hogwild or synchronous parallel SGD).
Page 10, "The hyperparameters of our approach...". What is the sensitivity of the experimental results to these hyperparameters? Moreover, how sensitive are the results to initialization?
Page 10, "we found that each parameter update of our method required about 80% more wall-clock time than an SGD update". How much of this is attributed to the fact that the statistics are computed asynchronously?
Pages 10-12, Experiments. There are no error bars in any of the plots, so it is impossible to ascertain the statistical significance of any of these results.
Page 11: Figure 2. Where is the Adam batchsize 50 line in the left plot? Why did the Adam batchsize 200 line disappear halfway through the right plot?

---

> ### Author Response · Authors · 2017-12-30
> **Response to AnonReviewer3**
>
> Thank you for your detailed comments.  We will address each of your major points in the sections below, followed by your remaining questions/comments.
>
>
> Empirical / theoretical analysis of approximation quality
> =========================
>
> A detailed discussion, empirical study, and analysis of the main approximation assumption (independence of the a’s and g’s) used in K-FAC and its derivatives (including this work) is contained in the original K-FAC paper.  However, these are not approximations bounds in the sense you likely mean.
>
> Due to the mathematically intractable nature of neural networks it is almost certainly impossible to provide such theoretical bounds.  Moreover, for each of these approximating assumptions, it is quite likely that there exists some artificially constructed model and dataset pair where they would be strongly violated.  And they are almost surely violated for real models and datasets as well, just to a lesser degree.
>
> An empirical study of each of these approximations would be interesting, but there are very many of them and so this would be a large undertaking.  We felt that this was outside of the scope of a conference paper, especially given that our manuscript was already on the long side.  Instead, we decided to evaluate the quality of our approximations in the only sense in which matters in practice: how well they translate into optimization performance on real tasks.
>
>
> RE prioritization of tractability over approximation quality
> ========================
>
> Approximations are often made for the sake of tractability, with little justification beyond how well their associated algorithms perform in practice.  For example, the diagonal approximations that drive most practical second-order optimization schemes (like Adam/RMSprop) are made purely for practical reasons, with very little theoretical or empirical justification.  And as one can see from the figures in the original K-FAC paper (see Figure 2 of Martens & Grosse, 2015), the true curvature matrix is highly non-diagonal in neural networks, so the diagonal approximation is indeed quite severe/inaccurate.
>
> While the approximations proposed in our paper are greater in sheer number, their sum total is still far less severe than a diagonal approximation.  (Diagonal approximations take the components of the gradient to be independent, thereby completely giving up on trying to model its inter-component statistical structure.)
>
> In designing our approximations we looked for the mildest possible ones that preserved the tractability properties we need.  Ultimately, tractability has to be the overriding consideration when designing algorithms that can be used in practice.   In one of your points you question our use of a linear-Gaussian model to describe the dependencies between the w_t’s.  However, the only obvious alternative to this, which would preserve tractability (in the context of the other approximations being made), is to neglect the dependencies between the w_t’s, thus treating them as statistically independent.  Should we give up on modeling these dependencies simply because the only tractable model we are aware of has no theoretical guarantees in terms of its accuracy?
>
> Perhaps better tractable approximations exist and could be the subject of future research.  Indeed, we can’t prove that they don’t exist, and would be excited to learn about them.  However, we feel that the onus should not be on us to provide such a proof.  Our contribution is a constructive existence proof of a non-obvious approximation to the Fisher of an RNN, which is a) much less severe that existing approaches (e.g. diagonal approximations), is b) validated on real data, and is c) useful in practice.  It feels like this should be good enough for a conference paper.
>
> Another very important point to keep in mind is that the approximations need not be particularly accurate for them to be useful in optimization.  Consider the analogy to statistical data modeling.  People frequently use primitive statistical models (e.g. linear-Gaussian) to describe data distributions that their models cannot possibly ever capture faithfully.  Nonetheless, these models have some predictive power and utility insofar as there are some aspects of the true underlying data-generated processes that can be described (however approximately) by such a simple model.  Our situation is analogous.  Our approximations, while they are clearly imperfect, allow us to capture enough of the statistical structure of the gradients that the resulting approximate Fisher still has some utility.  They could be wildly inaccurate in absolute terms and still be useful for our purposes.
>
> (continued in the next reply)

---

> > ### Author Response · Authors · 2017-12-30
> > **Response to AnonReviewer3 (continued)**
> >
> > RE Intuitive justifications for approximations
> > ======================
> >
> > Several of the key approximations we used were given intuitive justifications.  For example, we justified the use of a chain-structure model for the w_t’s by pointing at that they are produced by a process (forward evaluation followed by back-prop) that has a similar sequential chain structure.  We also provided intuitive justification and some preliminary analysis for Option 2.
> >
> > However several of the approximations were not given intuitive justifications, as you point out, and so we will add the following snippets of text to the respective sections.
> >
> > - Independence of T and the w_t's is a reasonable approximation assumption to make because 1) for many datasets T is constant (which formally implies independence), and 2) even when T varies substantially, shorter sequences will typically have similar statistical properties to longer ones (e.g. short paragraphs of text versus longer paragraphs).
> >
> > - Temporal homogeneity is a pretty mild approximation, and is analogous to the frequently used “steady-state assumption” from dynamical systems.  Essentially, it is the assumption that the Markov chain defined by the system ``mixes" and reaches its equilibrium distribution.  If the system has any randomness, and its inputs reach steady-state, the steady-state assumption is quite accurate for states sufficiently far from the beginning of the sequence (which will be most of them).
> >
> > - V1 is symmetric iff \hat{\Psi} is symmetric.  And as shown in the proof of Proposition 1 (see Appendix A.1) \hat{\Psi} has the interpretation of being the transition matrix of an LGGM which describes the evolution of “whitened” versions of the w_t’s (given by \hat{w_t} = V_0^{-1/2} w_t).  Linear dynamical systems with symmetric transition matrices arise frequently in machine learning and related areas (Huang et al., 2016; Hazan et al., 2017), particularly because of the algorithmic techniques they enable. Intuitively, a symmetric transition matrix allows allows one to model exponential decay of different basis components of the signal over time, but not rotations between these components (which are required to model sinusoidal/oscillating signals).
> >
> > Huang, Wenbing, et al. "Sparse coding and dictionary learning with linear dynamical systems." Proceedings of the IEEE Conference on Computer Vision and Pattern Recognition. 2016.
> >
> > Hazan, Elad, Karan Singh, and Cyril Zhang. "Learning linear dynamical systems via spectral filtering." Advances in Neural Information Processing Systems. 2017.
> >
> > (continued in next reply)

---

> > > ### Author Response · Authors · 2017-12-30
> > > **Response to AnonReviewer3 (continued)**
> > >
> > > Answers to specific questions not addressed above:
> > > =======================
> > >
> > > Page 2, “DV”:   V is a free variable used to define the D[…] notation.  We have added a clarification of this in the text.
> > >
> > > Page 2, “p_{y|x}(\theta)”:  p is defined in relation to f and L via the equation -log p(y|x,\theta) = -log r (y|f(x,\theta)) = L(y, f(x,\theta) near the start of Section 2.1.  We will get rid of the “p” notation to simplify things.
> > >
> > > Page 3, "common practice…”: Sorry, this was a typo.  It should have read "(F + \lambda I)^{-1} \nabla h instead of F^{-1} \nabla h”
> > >
> > > Page 3, “approximate g and a”:  This is the central approximation of the K-FAC approach and is discussed in the original paper (see Section 3.1 of Martens & Grosse [2015]).  It is shown that the approximation is equivalent to neglecting the higher-order cumulants of the a’s and g’s, or equivalently assuming that they are Gaussian distributed.  We will add a sentence or two pointing this out.
> > >
> > > Again though, this justification is primarily a statistical way to interpret an approximation that is made for the sake of algorithmic tractability.  It seems likely that it could violated to an arbitrarily large degree by specially constructed examples, which is a possibility that Martens & Grosse acknowledge in the original K-FAC paper.  Despite this, it works well enough in practice to be a good alternative to diagonal approximations.
> > >
> > > Page 7, “But insofar as the w_t’s…”: This is saying that a process with hidden state will behave in an approximately Markovian way if the observed state contains most of the information of the hidden state.  (If it contains *all* of the information of the hidden state then it is exactly Markovian.)
> > >
> > > Page 8, “Option 2…”:   There is no single sequence length which will make this approximation accurate in practice.  It will strongly depend on how close the temporal autocorrelation is to 1.
> > >
> > > The expression measures error in the eigenvalues.  This can be translated back to a bound on the error in F (induced by this particular approximation) by just pre and post-multiplying by U and U^\top respectively.  But this doesn’t actually do anything and just results in a more cluttered expression that is harder to interpret.
> > >
> > > The purpose of this analysis was merely to establish the nature of the relationship between T, the temporal autocorrelations, and the approximation error (due to this particular part of the approximation), up to a proportionality constant.
> > >
> > > Page 9, “The 2nd-order statistics…”:  We used the same setting of 0.95 for the decay constant in all of our experiments.  This was the same value used in the previous papers on K-FAC.
> > >
> > > Page 10, “The additional computations…”:  For large RNN like the ones we trained in our experiments, computing gradients on the CPU tends to be about 3-4 times slower than on the GPU.  Thus we suspect that using the extra CPU resources for gradient computations would have provided only marginal improvement to SGD, especially when one accounts for the fact that SGD benefits considerably less from using larger minibatches than K-FAC does.
> > >
> > > Also, the reason we performed the inverse computations on the CPUs for K-FAC is that that the GPU implementation of the required matrix-decompositions operations (inverse, SVD, eigen-decomposition, etc) are surprisingly slow compared to the CPU.  This may be due to the more serial nature of such computations.  We weren’t trying to give K-FAC an unfair advantage.
> > >
> > > Page 10, “The hyperparameters…”:  Good settings of the hyperparameters are crucial for good performance, for both our method and baselines we compared to.  The results are not particularly sensitive to the exact values, since performance varies as a relatively smooth continuous function of the hyperparameter settings (as it does with any reasonable method.)
> > >
> > > Likewise the networks must be initialized carefully for *any* of the optimization methods to do well.  (Each method used the same initialization in our experiments.)
> > >
> > > Page 10-12, “Experiments.”  Error bars are almost never included for optimization benchmarks of standard supervised neural network training.  This is because the training curves tend to be very predictable and well behaved across multiple seeds.  This is different from the situation in deep reinforcement learning for example, where the random and sparse nature of exploration introduces a lot of variability.
> > >
> > > Page 11, “Figure 2”:  Sorry about this. Because Adam batch size 50 is about 4 times slower than the batch size 200 in terms of the per-update progress. That is why we do not see the Adam batch size 50 in the left plot is. The black line gets to 1.4 bits-per-character after 22,000 updates. And, the experiments for batch size 200 terminated early because we used the same total number of updates for each configuration.  We will update the figures in the next revision with an extended run.

---

### Official Review · AnonReviewer1 · 2017-11-27
**Nice extension of K-FAC for RNNs, but missing validation/testing performances**

**Rating:** 7
**Confidence:** 4

**Review:**


Summary of the paper
-------------------------------

The authors extend the K-FAC method to RNNs. Due to the nature of BPTT, the approximation that the activations 'a' are independent from the gradients 'Ds' doesn't hold anymore and thus other approximations have to be made. They present 3 ways of approximating F, and show optimization results on 3 datasets, outperforming ADAM in both number of updates and computation time.

Clarity, Significance and Correctness
--------------------------------------------------

Clarity: Above average. The mathematical notations is overly verbose, so it makes the paper harder to understand. Note that it is not the author's fault only, since they followed the notation of the other K-FAC papers.
For instance, the notations goes from 'E[xy^T]' to 'cov(x, y)' to 'V_{x,y}'. I don't think introducing the 'cov' notation helps with the understanding of the paper (unless they explicitly wanted to stress out that the covariance of the gradients of the outputs of the model are centered). Also the 'V' in equation (4) could be confused with the 'V' in the first equation. Moreover, for the gradients with respect to the activations, we go from 'dL/ds' to 'Ds' to 'g', and for the weights we go from 'dL/dW' to 'DW' to 'w'. Why not keeping the 'Ds' and 'Dw' notation throughout the paper, and defining Dx as vec(dL/dx)?

Significance: This paper aims at helping with the optimization of RNNs and is thus and important contribution for our community.

Correctness: The paper is technically correct.

Questions
--------------

1. In figure 1, how does it compare to Adam instead of SGD? I think it would be a more fair comparison since SGD is rarely used to train RNNs (as RMSprop and ADAM might help with the vanishing/exploding gradients problem). Also, does the SGD baseline has momentum (since your method does)?
2. In all experiments, how do the validation / testing curves look like?
3. How does it compare to different reparametrization techniques, such as Layer Normalization or Batch Normalization?

Pros
------

1. This paper completes the K-FAC family.
2. It addresses the optimization of RNNs, which is an important research direction in our field.
3. It shows different levels of approximations of the Fisher, with the corresponding performances.

Cons
-------

1. No validation / test curves for any experiments, which makes it hard to asses if one should use this method in practice or not.
2. The notation is a bit verbose and can become confusing.
3. Small experimental setting (only PTB and DNC).

Typos
--------

1. Sec 1, par 5: "a new family curvature" -> "a new family of curvature"

---

> ### Author Response · Authors · 2017-12-31
> **Response to AnonReviewer1**
>
> Thank you for your detailed review.  See below for our response to your various questions and concerns.
>
> Notation
> -----------
>
> The “V” in the first equation is just a free/arbitrary variable to define the D[...] notation.  We will use a different symbol to avoid confusion, and clarify that it’s a free variable in the first equation.
>
> Because the zero-centered nature of the second-order statistics are not crucial we will replace the use of covariances with uncentered 2nd-order moments, as you suggest.
>
> We never actually use the notation dL/dZ anywhere except to define the DZ notation.  Because of the way that gradient quantities appear in complex expressions in our paper (often multiple times in the same expression), this shortened notation seems necessary to avoid producing very long and ugly expressions that are hard to parse.  Unfortunately, it is not feasible to define DZ = vec(dL/DZ), since we need to use the non-vectorized version at different points.
>
>
> Questions
> ------------
>
> 1.
>
> SGD has become the goto optimizer for these PTB tasks due to its superior generalization properties (Merity et al, 2017; Wilson et al. 2017), which is why we used it in our experiments.  But since our paper is not concerned with generalization (see our answer to your second question below) there is a good argument for using Adam as a second baseline, so we will include this in an upcoming revision of the manuscript.
>
> Also, we would observe that diagonal methods like RMSProp / Adam likely won’t do anything to address the vanishing or exploding gradients problem in RNNs (as suggested in your comment).  This is because the parameters are shared across time, and contributions from all the time-steps are added together before any preconditioning is applied.
>
> The same argument also applies to a non-diagonal method like K-FAC.  However, if the gradient contributions from different time-steps happen to land inside of distinct subspaces of the parameter space, then a non-diagonal method like K-FAC may still help with vanishing/exploding gradients by individually rescaling each of these contributions.  (See Section 3.3.1 of J Martens’ thesis http://www.cs.toronto.edu/~jmartens/docs/thesis_phd_martens.pdf).
>
> Merity, Stephen, Nitish Shirish Keskar, and Richard Socher. "Regularizing and optimizing LSTM language models." arXiv preprint arXiv:1708.02182 (2017).
>
> Wilson, Ashia C., et al. "The Marginal Value of Adaptive Gradient Methods in Machine Learning." arXiv preprint arXiv:1705.08292 (2017).
>
>
> 2.
>
> In our experiments K-FAC did overfit more than SGD.  The final perplexity values on the PTB tasks were about 5-8 points higher.  Please see Appendix D in the latest revision for the generalization performance.
>
> The reasons why we didn’t present test performance in the existing version of the paper, and why we stand by this decision, are discussed below.
>
> The tendency for SGD w/ early-stopping to self-regularize is well-documented, and there are many compelling theories about why this happens (e.g. Duvenaud et al., 2016 ; Hardt et al, 2015).  It is also well-known that 2nd-order methods, including K-FAC and diagonal methods like Adam/RMSprop, don’t self-regularize nearly as much (e.g. Wilson et al, 2017; Keskar et al, 2017).
>
> But just because a method like K-FAC doesn’t self-regularize as much as SGD, this doesn’t mean that it isn’t of practical utility. (Otherwise diagonal 2nd-order methods like Adam and RMSprop wouldn’t be as widely used as they are.)  Implicit self-regularization of the form that SGD has can always be replaced by *explicit* regularization (i.e. modification of the loss) and/or model modifications.  Moreover, in the online or large-data setting, where each example is processed only once, there is no question of generalization gap because the population loss is directly (stochastically) optimized. This online setting is encountered frequently in language modeling tasks, and our K-FAC method is particularly relevant for such tasks.
>
> (continued in next reply)

---

> > ### Author Response · Authors · 2017-12-31
> > **Response to AnonReviewer1 (continued)**
> >
> > The key distinction to understand here is the difference between "optimization benchmarks" and "learning benchmarks".
> >
> > An optimization benchmark is concerned with the rate of empirical loss minimization, i.e. optimization performance.  For an optimization benchmark to be valid it must use the same objective function for each optimizer.  It is, by definition, not concerned with performance on any other objective than the one which is being optimized.  (In some sense it cannot be without becoming an incoherent concept.)
> >
> > Learning benchmarks, meanwhile, test how well one can train a model that generalizes to the test set.  Performance is measured using the test loss.  In such benchmarks, the optimizer, the regularization, and even details of the model itself, can all be varied.
> >
> > To properly assess the usefulness of K-FAC within machine learning we would have to run a comprehensive learning benchmark where, given a fixed dataset, the regularization (and possibly the model too) could be tuned for each optimizer.  Due to the known interaction between optimization methods and regularization, the need to do this optimizer-specific tuning seems unavoidable for the test to be fair.  Moreover, the standard models and regularization configs that we use in our experiments were already tuned (by their original authors) to give good generalization performance with SGD.
> >
> > Simply looking at the test curves after running an optimization benchmark is a very poor substitute for a proper learning benchmark.  Indeed, it seems impossible to design a single experiment that can simultaneously function as both an optimization and learning benchmark, since the former requires the use of a fixed objective function, while the latter needs the objective to be varied.
> >
> > Because this paper is about optimization we stuck to optimization benchmarks.  While a comprehensive learning benchmark would certainly be valuable (not just to assess the usefulness of K-FAC, but other 2nd-order methods as well), we believe it is out of scope for this work.
> >
> >
> > Duvenaud, David, Dougal Maclaurin, and Ryan Adams. "Early stopping as nonparametric variational inference." Artificial Intelligence and Statistics. 2016.
> >
> > Hardt, Moritz, Benjamin Recht, and Yoram Singer. "Train faster, generalize better: Stability of stochastic gradient descent." arXiv preprint arXiv:1509.01240 (2015).
> >
> > Keskar, Nitish Shirish, and Richard Socher. "Improving Generalization Performance by Switching from Adam to SGD." arXiv preprint arXiv:1712.07628 (2017).
> >
> > Wilson, Ashia C., et al. "The Marginal Value of Adaptive Gradient Methods in Machine Learning." arXiv preprint arXiv:1705.08292 (2017).
> >
> >
> > 3.
> >
> > In our latest revision we have included additional benchmark experiments suggested by the reviewer with the Adam optimizer and layer-normalization. While Adam outperforms SGD in the first few epochs, SGD obtains a lower loss at the end of training. We found layer-normalization helps speed up Adam considerably, but it hurts the SGD performance. Such an observation is consistent with previous findings.  In comparison, our proposed method significantly outperform both the Adam and the SGD baselines even with the help of layer-normalization.

---

> > > ### Comment · AnonReviewer1 · 2018-01-11
> > > **Thanks for your detailed answer!**
> > >
> > >
> > > Notation:
> > >
> > > Thanks for changing the notation, it is clearer in my opinion.
> > > "Unfortunately, it is not feasible to define DZ = vec(dL/DZ)": Oh right, sorry!
> > >
> > > Questions:
> > >
> > > 1. Thanks for the references, I definitively need to take a look. With the Adam curve, it is now clear that SGD is (or was) indeed the optimizer to use on this dataset.
> > >
> > > 2. I totally agree with you and really like your distinction between "optimization benchmarks" and "learning benchmarks". However, I still think that adding the validation curves (as you did) is quite useful for people willing to use your method in a learning benchmark setup. Even if K-FAC might require a bit more regularization than the traditional SGD, it might still provide them some gains in training speed.
> > >
> > > 3. It is also really nice to see that K-FAC works better than SGD with layer normalization. It is definitively a good argument in favor of K-FAC.
> > >
> > > Given the elements added to the paper and the insightful answers to my questions, I will change my grade for the paper.

---

### Official Review · AnonReviewer2 · 2017-11-27
**Novel method, incremental but sufficient improvements**

**Rating:** 7
**Confidence:** 3

**Review:**

In this paper, the authors present a second-order method that is specifically designed for RNNs. The paper overall is well-written and I enjoyed reading the paper.

The main idea of the paper is to extend the existing kronecker-factored algorithms to RNNs. In order to obtain a tractable formulation, the authors impose certain assumptions and provide detailed derivations. Even though the gain in the convergence speed is not very impressive and the algorithm is quite complicated and possibly not very accessible by deep learning practitioners, I still believe this is a novel and valuable contribution and will be of interest to the community.

I only have some minor corrections:

1) Sec 2.1: typo "is is"
2) Sec 2.2: typo "esstentiallybe"
3) Sec 2.2: (F+lambda I) --> should be inverse
4) The authors should include a proper conclusion

---

> ### Author Response · Authors · 2017-12-30
> **Response to AnonReviewer2**
>
> Thanks for your comments.  We have corrected the errors you pointed out and added back in the conclusion, which was originally cut for space considerations.  These will appear in our next revision (to be posted soon).
>
> With regards to convergence speed, we feel that the gains over SGD/Adam are significant.   While wall-clock time wasn’t improved substantially in the DNC experiment (Figure 3), it was on the first two experiments on Penn-TreeBank (Figures 1 and 2).  From those latter two figures one can clearly see that SGD/Adam slow down at a significantly higher loss than our method (almost to the point of plateauing).
>
> While we agree that the method is challenging to implement, we have a TensorFlow implementation ready for public release.  We will make this available as soon as we can while respecting the anonymity of the reviewing process.

---

### Author Response · Authors · 2017-11-23
**Typo found in Section 3.5.4**

We found a typo in Section 3.5.4 which may confuse the reviewers.

Near the top of that section the equation should be:

V_1 = V_{1, 0} = cov ( w_1, w_0) = cov ( \Psi w_0 + \epsilon_1, w_0) = \Psi cov ( w_0, w_0) + cov ( \epsilon_1, w_0) = \Psi V_0 + 0 = \Psi V_0.

Sorry for any confusion this may have caused.

---

### Author Response · Authors · 2017-12-31
**Paper updated**

We have updated the paper based on the suggestions of the reviewers.

This revision doesn't contain any of the planned updates/changes to the experimental results.  These will come in the next revision.

---

### Author Response · Authors · 2018-01-01
**Paper updated again**

We've updated the paper again with the new experimental data (test scores, comparisons to layer-norm, etc).

We've also edited the review responses to reflect these updates.

---

### Decision · Program_Chairs · 2018-01-29
**ICLR 2018 Conference Acceptance Decision**

**Decision:**

Accept (Poster)

**Comment:**

This clearly written paper extends the Kronecker-factored approximate curvature optimizer to recurrent networks.  Experiments on Penn Treebank language modeling and training of differentiable neural computers on a repeated copy task show that the proposed K-FAC optimizers are stronger than SGD, Adam, and Adam with layer normalization. The most negative reviewer objected to a lack of theoretical error bounds on the approximations made, but the authors successfully argue that obtaining such bounds would require making assumptions that are likely to be violated in practice, and that strong empirical performance on real tasks is sufficient justification for the approximations.

Pros:
+ "Completes" K-FAC training by extending it to recurrent models.
+ Experiments show effects of different K-FAC approximations.

Cons:
- The algorithm is rather complex to implement.